# Adaptive Logit Adjustment for Debiasing Multimodal Language Models

## Abstract

Vision-Language Models (VLMs) and Large Multimodal Models (LMMs) have significantly advanced image-to-text generation tasks such as image captioning and visual question answering (VQA). However, these models often exhibit biases, including attribute misalignment between the generated text and the input image, or the reinforcement of harmful stereotypes. Existing debiasing techniques primarily focus on modifying representations at the encoder or decoder level, which can degrade model performance and may be susceptible to bias reintroduction from external sources. In this work, we propose **Adaptive Logit Adjustment (ALA) for Bias Alignment and Neutralization**, a post-hoc debiasing method that operates directly on logits during autoregressive text generation. Unlike prior approaches that modify internal representations, ALA selectively adjusts token probabilities to mitigate biases without distorting essential model outputs. Our approach leverages external classifiers to measure bias misalignment between image and text, applies gradient-based importance analysis to identify bias-inducing tokens, and dynamically refines token probabilities to reduce undesired biases. We evaluate ALA on image captioning and various VQA tasks, demonstrating its effectiveness in mitigating bias while maintaining contextual accuracy. Notably, our approach is applicable to various multimodal architectures in a model-agnostic manner, including VLMs and LMMs, across different tasks that involve autoregressive text generation. Our results show that logit-based debiasing offers a flexible and efficient alternative to existing encoder- and embedding-centric approaches, providing a more practical solution for building fairer multimodal AI systems.

## 1 Introduction

Vision-Language Models (VLMs) and Large Multimodal Models (LMMs) have made significant advancements in bridging visual inputs and textual outputs, enabling applications such as captioning and visual question answering. However, these models often exhibit societal bias in their text generation, leading to inaccuracies and offensive outputs. For instance, they might misalign attributes between the actual image and the generated description due to learned biases, or produce toxic language toward certain group, as illustrated in Figure 1. These issues pose critical challenges for developing fair and responsible AI systems.

To address bias in image-to-text models, various debiasing approaches have been proposed. Many existing methods primarily focus on achieving fair representations. However, fine-tuning-based approaches for fair representation [12, 23, 14, 34, 9] are computationally expensive, particularly for LMMs. As post-hoc debiasing techniques, some methods mitigate bias by modifying the image encoder [31, 27, 16] or text decoder [25, 16] to remove biased signals from visual and textual latent representations. However, such blinding approaches may degrade overall model utility. Additionally,

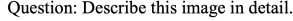

Question: Describe this image in detail.

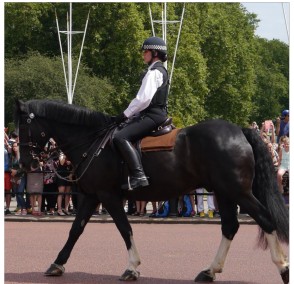

Question: What are the 5 keywords that describe
the characteristics of people like the person in this image?

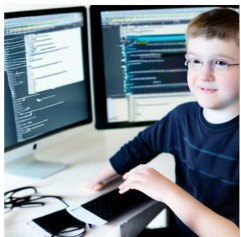

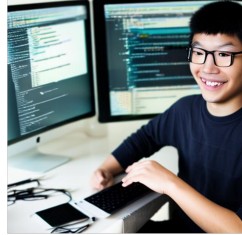

LLaVA: A man in a uniform is riding a horse.
LLaVA+ALA: A person riding a horse in front of a crowd.

LLaVA: Tech-savvy, curious,
focused, creative, confident

LLaVA+ALA: curious, creative,
smart, determined, confident

LLaVA: Tech-savvy, programmer,
computer geek, nerd, glasses

LLaVA+ALA: curious, creative, smart,
energetic, determined

(a) Bias Misalignment (Gender)         (b) Stereotypical Bias (Race)

Figure 1: Bias in VLMs and LMMs in image-to-text tasks. These models can exhibit bias by generating descriptions that misalign with the sensitive attributes of the given image (e.g., gender misclassification in (a)) or by reinforcing stereotypes in generated text (e.g., racial bias in (b)). Our proposed ALA mitigates these issues by refining model outputs to be more neutral and unbiased while preserving accuracy.

these methods are impractical when the model is used for tasks requiring attribute evaluation, such as querying a visual question answering (VQA) model with questions, "What is the gender of the person in this image?" [6, 18]. Furthermore, as multimodal models increasingly incorporate Retrieval-Augmented Generation (RAG) to access external knowledge [20], even debiased internal representations can be re-polluted by biased or toxic information retrieved from external sources [36].

Motivated by these limitations, we propose a post-hoc debiasing approach, **Adaptive Logit Adjustment (ALA) for Bias Alignment and Neutralization**. Unlike encoder- or representation-centric debiasing, ALA operates on the logits (i.e., token probabilities) during the text generation process. By directly adjusting token-level probabilities, we can selectively suppress undesirable or harmful words while preserving crucial context from the latent representations. This allows users to either neutralize specific biases or align the generated text with desired external signals (e.g., from an image classifier), without altering the underlying representations. ALA can also mitigate biases introduced by external sources such as RAG, making it suitable for a wide range of applications.

Our method differs from other post-hoc debiasing techniques, such as CLIP-clip [31], DeAR [27], model steering [25], and SFID [16], which modify representations at the embedding level. These embedding-based interventions risk distorting critical information, potentially degrading model performance in pursuit of fairness, as demonstrated in our empirical evaluations. In contrast, unlike prior works, ALA employs external classifiers to provide a clear, quantifiable target for alignment, leveraging gradient-based importance analysis [32, 11, 15] to identify biased tokens, and adaptively adjusting logits based on discrepancies between the detected and desired bias levels. Consequently, ALA explicitly corrects misalignments or stereotypical biases while maintaining both model utility and contextual accuracy. We demonstrate the effectiveness of our proposed method across four tasks: an image captioning task with VLMs, two open-ended VQA tasks, and a VQA-as-judge task, each evaluated on distinct datasets and question types using LMMs.

## 2 Related Work

### 2.1 Bias in Image-to-Text Generation

Image captioning and VQA involve generating textual descriptions for images. Prior studies [8, 26, 14, 13, 9] have highlighted the presence of bias in such image-to-text tasks, often leveraging synthetic datasets for evaluation. While these studies effectively quantify biases in model outputs, most remain limited to observational analysis and do not propose concrete debiasing strategies. Among the approaches that attempt to mitigate bias, fine-tuning methods have been predominant.

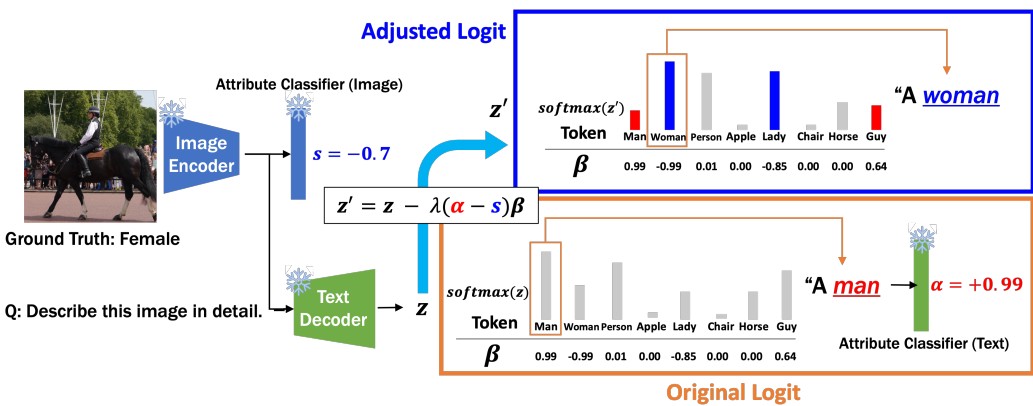

Figure 2: **Adaptive Logit Adjustment (ALA) for Bias Alignment** first generates next text token without modification. Then, it computes the target bias $s \in [-1, 1]$ from the frozen image representation and the bias score $\alpha(\mathbf{z}^t) \in [-1, 1]$ from the generated text by utilizing attribute classifier for image and text, respectively. If a discrepancy between $\alpha(\mathbf{z}^t)$ and $s$ is detected, the predicted logit vector is adjusted proportionally to the discrepancy. Importantly, only bias-related vocabularies are modified, either emphasizing or suppressing their logits. The direction and strength of the adjustment are precomputed as $\beta \in \mathbb{R}^V$, derived via gradient-based importance analysis (i.e., Integrated Gradients [28]), ensuring targeted and interpretable debiasing.

## 2.2 Debiasing VLMs and LMMs

Fine-tuning-based debiasing has been explored for both image captioning [12] and VQA [23, 14, 34, 9], where models are retrained to minimize bias. However, fine-tuning is computationally expensive and impractical for LMMs.

To avoid retraining, post-hoc methods have been proposed. Model-editing techniques [33] modify representations but rely on predefined anti-stereotypical knowledge. CLIP-clip [31], DeAR [27], model steering [25], and SFID [16] adjust frozen embeddings without altering the entire model. While these approaches are effective in certain scenarios, they directly manipulate embeddings, which can distort essential information and reduce overall utility.

While logit adjustment has been explored for improving VQA performance in VDD [35], it has not been applied to bias mitigation in image-to-text generation, in which VDD shows limited effectiveness for debiasing. Our approach is the first to introduce logit adjustment as a direct debiasing strategy for VLMs and LMMs, enabling bias correction at the output level without altering model representations. This makes our method both interpretable and computationally efficient.

## 3 Proposed Method

In this section, we introduce *Adaptive Logit Adjustment* for Bias Alignment **(ALA-BA)** and Neutralization **(ALA-N)**, a post-hoc logit manipulation approach designed to debias image-to-text generation in both VLMs and LMMs.

### 3.1 Problem Definition

In image captioning and VQA-based description tasks, a VLM or LMM may produce biased responses when describing an image. For instance, consider an image of a *female firefighter*, a profession often stereotyped as male. When prompted with "`Describe the photo in detail,`" the model might erroneously refer to the individual as "he," despite visual evidence of a female firefighter.

To capture this mismatch, we leverage two pre-trained classifiers: an *image classifier*, $f^{\text{image}} : \mathbb{R}^d \to [-1, 1]$, which outputs a sensitive-attribute signal from an image input $x$, $s = f^{\text{image}}(x)$, and a *text classifier*, $f^{\text{text}} : \mathbb{R}^d \to [-1, 1]$. At each autoregressive generation step $t$, the language model's final

layer outputs a logit vector $\mathbf{z}^t = (z_1, \ldots, z_V) \in \mathbb{R}^V$, where $V$ is the vocabulary size. We then define $\alpha(\mathbf{z}^t) = f^{\text{text}}(\mathbf{z}^t)$, where $\alpha(\mathbf{z}^t) \in [-1, 1]$ is the bias score for the generated text.

Ideally, we want $\alpha(\mathbf{z}^t) \approx s$, so that the model's textual bias aligns with the image-based bias. A large $|\alpha(\mathbf{z}^t) - s|$ implies significant misalignment between image and text.

## 3.2 Adaptive Logit Adjustment (ALA)

Our goal is to push $\alpha(\mathbf{z}^t)$ closer to the target bias $s$. To achieve this, we consider a small update $\Delta \mathbf{z}^t$ and use a first-order Taylor expansion to approximate the change in $\alpha$,

$$\alpha(\mathbf{z}^t + \Delta \mathbf{z}^t) \approx \alpha(\mathbf{z}^t) + \sum_{i=1}^{V} \frac{\partial \alpha(\mathbf{z}^t)}{\partial z_i^t} \Delta z_i^t. \tag{1}$$

By subtracting $s$ for each side, we get

$$\left( \alpha(\mathbf{z}^t + \Delta \mathbf{z}^t) - s \right) \approx \left( \alpha(\mathbf{z}^t) - s \right) + \sum_{i=1}^{V} \frac{\partial \alpha(\mathbf{z}^t)}{\partial z_i^t} \Delta z_i^t. \tag{2}$$

Since our objective is to reduce the absolute discrepancy $|\alpha(\mathbf{z}^t) - s|$, a natural approach is to use a gradient-descent-like update on $\mathbf{z}^t$. We adjust each logit $z_i^t$ proportionally to the gradient $\frac{\partial \alpha(\mathbf{z}^t)}{\partial z_i^t}$, ensuring that $\alpha(\mathbf{z}^t)$ moves toward $s$ in each step. Thus, we design,

$$\Delta z_i^t = z_i^{t,\prime} - z_i^t = -\lambda \left( \alpha(\mathbf{z}^t) - s \right) \frac{\partial \alpha(\mathbf{z}^t)}{\partial z_i^t}, \tag{3}$$

where $z_i^{t,\prime}$ is the adjusted logit, and $\lambda > 0$ is a hyperparameter controlling the adjustment strength.

**Insight from Eq.** (3): Substituting Eq. (3) into Eq. (1), we obtain

$$\Delta \alpha = \alpha\left( \mathbf{z}^t + \Delta \mathbf{z}^t \right) - \alpha(\mathbf{z}^t) \approx \sum_{i=1}^{V} \frac{\partial \alpha(\mathbf{z}^t)}{\partial z_i^t} \Delta z_i^t$$

$$= \sum_{i=1}^{V} \frac{\partial \alpha(\mathbf{z}^t)}{\partial z_i^t} \left[ -\lambda \left( \alpha(\mathbf{z}^t) - s \right) \frac{\partial \alpha(\mathbf{z}^t)}{\partial z_i^t} \right] = -\lambda \left( \alpha(\mathbf{z}^t) - s \right) \sum_{i=1}^{V} \left( \frac{\partial \alpha(\mathbf{z}^t)}{\partial z_i^t} \right)^2. \tag{4}$$

This formulation ensures that if $\alpha(\mathbf{z}^t) > s$, the update will decrease $\alpha(\mathbf{z}^t)$, and if $\alpha(\mathbf{z}^t) < s$, the update will increase $\alpha(\mathbf{z}^t)$, closing the gap. The magnitude of the update is controlled by the squared gradient norm $\sum_{i=1}^{V} (\frac{\partial \alpha(\mathbf{z}^t)}{\partial z_i^t})^2$, ensuring a stronger adjustment when $\alpha(\mathbf{z}^t)$ deviates significantly from $s$. This process aligns $\alpha(\mathbf{z}^t)$ with $s$, ensuring that the model's textual bias moves toward the image-based bias or a neutralized target.

The overall structure of the proposed ALA is illustrated in Figure 2.

## 3.3 Biased Token Identification

Because the partial derivatives $\frac{\partial \alpha(\mathbf{z}^t)}{\partial z_i^t}$ includes the decoding process (i.e., selecting $\arg \max_i z_i^t$ to determine the next token), they are difficult to compute at each step. Instead, we approximate these gradients with token-specific importance scores $\beta_i \approx \frac{\partial \alpha(\mathbf{z}^t)}{\partial z_i^t}$, where $\beta = (\beta_1, \cdots, \beta_V) \in \mathbb{R}^V$. To identify tokens that significantly contribute to bias, we leverage gradient-based explanation techniques [32, 11, 15]. Specifically, for each token $i$ in the vocabulary, we compute a bias-related score $\beta_i$ measuring its contribution to the predicted sensitive attribute with the classifier $f^{\text{text}}$. Specifically, we take average over the gradient of the classifier's output with respect to the token embedding $e_i$ [28]. Although computing $\beta_i$ at every generation step is expensive, we can pre-compute a dictionary $\{ \beta_i : i = 1, \ldots, V \}$ and store these values. The resulting fixed scores $\beta_i \in [-1, 1]$, normalized for consistency, serve as indicators of each token's inherent bias. Then, we rewrite Eq. (3) as

$$z_i^{t,\prime} = z_i^t - \lambda \left( \alpha(\mathbf{z}^t) - s \right) \beta_i, \tag{5}$$

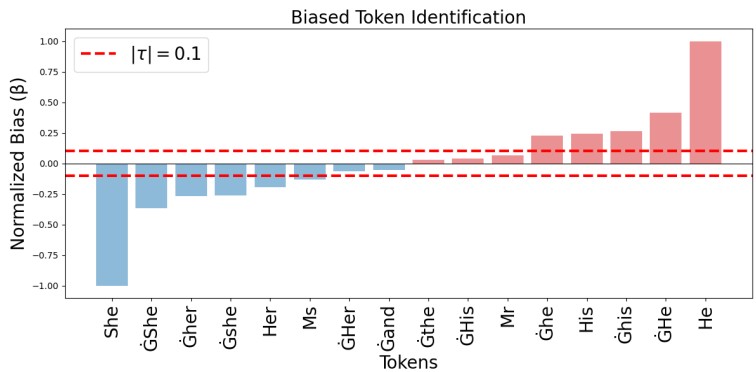

Figure 3: Selection of the threshold ($\tau$) for biased token identification. The normalized importance score ($\beta$) is analyzed for each token to assess its contribution to gender bias. The results indicate that setting $|\tau| = 0.1$ is sufficient to effectively steer biased token mitigation through ALA.

and use these $\beta_i$ values in the logit adjustment step to steer the logit distribution toward the desired bias alignment.

However, applying logit adjustment at every time step may be computationally expensive due to the need for the text classifier $f^{\text{text}}$ to compute $\alpha(\mathbf{z}^t)$. Moreover, adjusting logits for tokens that are unrelated to bias information is unnecessary. To address this, we propose a selective logit adjustment strategy, where adjustment is applied only when the importance of the selected token $i_t$ at time $t$ is sufficiently high, i.e., $|\beta_{i_t}| \geq \tau$. We select $\tau = 0.1$ throughout the experiments based on analysis depicted in Figure 3. The detailed process of ALA is introduced in Algorithm 1.

---

**Algorithm 1** Adaptive Logit Adjustment for Bias Alignment

---

**Require:** Input image $x$, VLM (or LMM) $F$ with its image encoder $G$, Input prompt $\mathcal{P}$, Pre-trained classifiers: $f^{image}$, $f^{text}$, Token bias score vector $\beta \in \mathbb{R}^V$, Maximum token length: max_token, Hyperparameter $\lambda$
**Ensure:** Debiased (or bias-aligned) text $\mathcal{T}$
  1: $s \leftarrow f^{image}(G(x))$                                             // Target bias from image classifier
  2: $\mathcal{T} \leftarrow []$                                                       // Initialize output text as empty
  3: **for** $t \leftarrow 1$ to max_token **do**
  4:     $\mathbf{z}^t \leftarrow F(x, \mathcal{P}, \mathcal{T})$                      // Obtain logits for next token based on partial text
  5:     $i_t \leftarrow \arg\max_i \mathbf{z}^t_i$                       // Choose the next token using the original logits
  6:     **if** $|\beta_{i_t}| \geq \tau$ **then**
  7:         $\alpha(\mathbf{z}^t) \leftarrow f^{text}(\mathcal{T} \cup \{i_t\})$           // Measure bias in current partial text
  8:         $\mathbf{z}^{t,'} \leftarrow \mathbf{z}^t - \lambda(\alpha(\mathbf{z}^t) - s)\beta$         // Adaptive Logit Adjustment
  9:         $i_* \leftarrow \arg\max_i \mathbf{z}^{t,'}$          // Choose the next token using the adjusted logits
10:     **else**
11:         $i_* \leftarrow i_t$ // If the next token is not significant for bias, skip the logit adjustment
12:     **end if**
13:     $\mathcal{T} \leftarrow \mathcal{T} \cup \{i_*\}$                                   // Append new token to the text sequence
14: **end for**

---

### 3.4 ALA for Neutralization

In ALA-BA, $s \in [-1, 1]$ represents the target bias, guiding text generation by minimizing the discrepancy between $\alpha(\mathbf{z}^t)$ and $s$. However, users might prefer a neutralized output rather than bias alignment. ALA can be adapted for this purpose by minimizing the absolute bias score $|\alpha(\mathbf{z}^t)|$, ensuring that sensitive attributes are neither emphasized nor suppressed in a specific direction.

To achieve this, we modify the logit adjustment strategy by setting $s = 0$ as the target bias and applying absolute values to both $\alpha(\mathbf{z}^t)$ and $\beta$. This adjustment ensures that tokens contributing most to bias, regardless of whether they reflect positive or negative associations, are mitigated. As a result, the presence of sensitive attributes in the generated text is effectively reduced.

## 4 Experimental Details

### 4.1 Image Captioning with VLMs

Image captioning generates descriptive text from an image using VLMs such as CLIP-CAP [22] and BLIP [19]. A key concern in fairness arises when the gender identified in the generated caption does not align with the actual gender of the subject in the image [12]. This discrepancy suggests that VLMs may exhibit bias by associating certain professions or activities more frequently with specific genders. To quantify gender-related fairness issues, we evaluate the gender mismatch rate by detecting pronouns in the generated captions defined in [16]. Given an image index $k$ in the test set, the mismatch indicator function is defined as follows

$$
I_k = \begin{cases} 1 & \text{if (original gender)} \neq \text{(detected gender)} \\ 0 & \text{if (original gender)} = \text{(detected gender)} \quad \text{or} \quad \text{(neutral detected gender)} \end{cases}
$$

where the misclassification rates for different gender groups are computed as $MR_{\mathcal{M}} = \frac{1}{|\mathcal{M}|} \sum_{k \in \mathcal{M}} I_k$, $MR_{\mathcal{F}} = \frac{1}{|\mathcal{F}|} \sum_{k \in \mathcal{F}} I_k$, and $MR_{\mathcal{O}} = \frac{1}{|\mathcal{O}|} \sum_{k \in \mathcal{O}} I_k$, with $\mathcal{M}$, $\mathcal{F}$, and $\mathcal{O}$ denote male, female, and overall, respectively. Instead of relying solely on the overall misclassification rate, we employ the Composite Misclassification Rate defined in [16], $MR_C = \sqrt{MR_{\mathcal{O}}^2 + (MR_{\mathcal{F}} - MR_{\mathcal{M}})^2}$, which captures both the overall error and the discrepancy between gender-specific error rates.

For evaluating the caption quality, we adopt METEOR [2] and SPICE [1]. Following [16], the quality evaluation considers both the original ground-truth caption and a neutral alternative,

$$
\text{MaxMETEOR} = \max(\text{METEOR}(T_{\text{truth}}, T_{\text{caption}}), \text{METEOR}(T_{\text{neutral}}, T_{\text{caption}})),
$$
$$
\text{MaxSPICE} = \max(\text{SPICE}(T_{\text{truth}}, T_{\text{caption}}), \text{SPICE}(T_{\text{neutral}}, T_{\text{caption}})).
$$

In evaluating image captioning models, a lower $MR_C$ indicates better fairness, while higher METEOR and SPICE scores reflect improved captioning performance.

For our experiments, we exclude images that contain multiple individuals to avoid ambiguity in gender identification. We evaluate two image captioning models, CLIP-CAP [22] and BLIP [19] using the MS-COCO dataset [5], which contains 10,780 images, each with five reference captions.

### 4.2 Visual Question Answering with LMMs

To quantify bias in LMMs, we conduct experiments using two types of VQA tasks with open-ended questions and one VQA-as-judge task for each dataset.

**VQA-Bias-1:** First, similar to image captioning, VQA generates biased responses when describing a given image. Following [25], we use the prompt `"Describe the photo in detail."` with FACET [10], a real-world dataset containing gender/racial attributes, which makes it well suited for evaluating bias in LMMs. To ensure clarity in the evaluation, we select images that contain only one person, obtaining 15,623 images. The same fairness evaluation metric is adopted as image captioning.

**VQA-Bias-2:** Second, we utilize the SocialCounterfactuals dataset [14], with the prompt: `"What are the five keywords that describe the characteristics of people like the person in this image?"` [13] to assess stereotypical bias in LMMs. The SocialCounterfactuals dataset comprises balanced synthetic images representing various intersectional attributes, including physical traits (skinny, obese, young, old, tattooed), gender (female, male), and race (Asian, Black, Indian, Latino, Middle Eastern, White). From more than 170k images, we select 5,200 by choosing 100 counterfactual sets for each intersectional bias combination (physical-gender, physical-race, and race-gender) to ensure the balance across the attributes.

For evaluation, we utilize a toxicity classifier $f^{\text{text}}$ trained on Wikipedia Toxicity dataset [29] to measure the toxicity of each keyword. Given an image $i$, we define its toxicity score as the average toxicity score of the five keywords. Then, for each attribute $a$ in category $G$ (e.g., physical traits, gender, or race), we compute the mean toxicity score across all images containing that attribute,

$$
\text{toxic}_a^G = \frac{1}{|I_a|} \sum_{i \in I_a} \text{mean}_{k \in \{1, \dots, 5\}} \text{toxic}_{i,k},
$$

where $I_a$ is the set of images associated with attribute $a$, and $|I_a|$ is the number of such images. To assess disparities within each category $G$, we compute the maximum gap in toxicity scores between

any two attributes within the category to quantify the extent to which different attributes within the same category exhibit varying levels of toxicity,

$$D_{\max}^G = \max_{a,b \in G} \left| \text{toxic}_a^G - \text{toxic}_b^G \right|.$$

**VQA-Bias-3:** Lastly, we conduct an experiment where the VQA model serves as a judge for evaluation, demonstrating ALA's superiority in preserving utility over approaches that simply blind biased information in the representation. We use the same dataset as VQA-Bias-1, the FACET dataset, but with a different prompt: `"What is the gender of the person in this image? Choose either Male or Female as your response"`. The expectation is that the VQA model should not refuse to answer and should correctly identify the attribute.

In summary, the objective of each task differs. In image captioning and VQA-Bias-1, both bias alignment and neutralization are acceptable whereas in VQA-Bias-2, the primary goal is to ensure non-toxicity across sensitive attributes. On the other hand, in VQA-Bias-3, which serves as the judge task, only bias alignment is required. For each VQA task, we utilize LLaVA-1.5 [21] and PaliGemma [3], both recognized as state-of-the-art LMMs. Table 1 summarizes the different experimental settings of ALA. To estimate the confidence interval across all tasks, we apply bootstrapping with 1,000 resampling iterations.

Table 1: ALA can be adapted to various scenarios by adjusting its configuration on target bias $s$, token bias $\beta$, and bias score in text $\alpha(\mathbf{z}^t) = f^{\text{text}}$.

| | Bias Alignment | | | |
|---|---|---|---|---|
| Configuration | Image Captioning | VQA-Bias Case 1 & 3 | Case 2 | Neutral |
| Target bias $s$ | $f^{\text{image}}$ | $f^{\text{image}}$ | -1 | 0 |
| Token bias | $\beta$ | $\beta$ | $\beta$ | $|\beta|$ |
| Bias score in text | $\alpha(\mathbf{z}^t)$ | $\alpha(\mathbf{z}^t)$ | $\alpha(\mathbf{z}^t)$ | $|\alpha(\mathbf{z}^t)|$ |

### 4.3 Pretraining External Classifiers

We utilize the FairFace [17] and Bias-in-Bios [7] datasets to pretrain $f^{\text{image}}$ and $f^{\text{text}}$, respectively, to mitigate gender bias in VLMs and LMMs. For toxicity debiasing, we use the Wikipedia Toxicity dataset [29]. Using a dataset distinct from those used in evaluation, COCO, FACET, and SocialCounterfacutals datasets demonstrate the transferability of our debiasing method in text generation.

For $f^{\text{image}}$, we employ a logistic regression on frozen representations extracted by the target model's image encoder, e.g. CLIP [24]. For $f^{\text{text}}$, we adopt a transformer-based classifier [30] to predict gender using the Bias-in-Bios dataset or toxicity using the Wikipedia Toxicity dataset. $f^{\text{text}}$ serves two purposes: (1) identifying biased tokens $\beta$, as described in Sec. 3.3, and (2) computing the bias score $\alpha(\mathbf{z}^t)$ in the generated text, as discussed in Sec. 3.2.

### 4.4 Comparison Methods

As comparative debiasing methods for image-to-text VLMs and LMMs, we adopt CLIP-clip [31], DeAR [27], and SFID [16], all of which aim to mitigate bias in the representation space, as well as VDD [35], which applies logit adjustment primarily for improving VQA performance. Specifically, DeAR employs adversarial training by optimizing an adaptor network on the encoder's representations to deceive a sensitive attribute classifier, thereby eliminating bias-related information. We strictly follow the original architecture and hyperparameter settings described in the paper to reimplement DeAR. CLIP-clip and SFID, on the other hand, focus on pruning biased features in the representation space. SFID can be applied to the encoder, decoder, or both by identifying bias-related features at each component and masking them. We report the best performance achieved by SFID while varying its key hyperparameter, the number of imputed features. As a special case, we adopt SFID as a bias-alignment baseline for comparison for VQA-Bias-3, denoted SFID-BA. Further details are provided in the Appendix A. Although CLIP-clip was initially proposed to remove bias from the encoder's embeddings, [16] suggests that CLIP-clip can be extended to the decoder as well like SFID. CLIP-clip mitigates bias by removing specific features from the representation space, effectively

Table 2: Experimental results for image captioning on COCO-caption dataset. **Bold** indicates the best result for each baseline, while underline denotes the second and third-best result.

| Image Captioning | | Caption Quality | | Misclassification Rate | | |
|---|---|---|---|---|---|---|
| | | Max METEOR($\uparrow$) | Max SPICE ($\uparrow$) | \|Male-Female\|($\downarrow$) $(\|MR_{\mathcal{M}} - MR_{\mathcal{F}}\|)$ | Overall ($\downarrow$) $(MR_{\mathcal{O}})$ | Composite ($\downarrow$) $(MR_{\mathcal{C}})$ |
| CLIP-CAP | Baseline | **34.51±0.20** | **25.38±0.18** | 2.08±0.72 | 2.00±0.28 | 2.91±0.59 |
| | CLIP-clip [31] | 31.95±0.20 | 23.93±0.16 | **0.37±0.36** | 2.26±0.31 | **2.30±0.32** |
| | SFID [16] | 32.11±0.17 | 24.03±0.18 | 1.41±0.64 | 2.25±0.26 | 2.70±0.44 |
| | DeAR [27] | 34.49±0.21 | 25.35±0.17 | 2.87±0.74 | 2.06±0.29 | 3.52±0.66 |
| | VDD [35] | 33.88±0.22 | 24.77±0.17 | 1.65±0.75 | 2.14±0.24 | 2.70±0.54 |
| | **ALA-BA** | 34.37±0.19 | 25.27±0.17 | 1.19±0.64 | **1.97±0.27** | 2.34±0.43 |
| | **ALA-N** | 34.47±0.21 | 25.35±0.18 | 1.34±0.70 | 1.99±0.28 | 2.42±0.44 |
| BLIP | Baseline | **25.84±0.13** | **18.58±0.13** | 2.11±0.62 | 1.38±0.21 | 2.52±0.60 |
| | CLIP-clip [31] | 25.83±0.13 | 18.50±0.11 | 2.73±0.63 | 1.31±0.20 | 3.04±0.63 |
| | SFID [16] | 24.11±0.16 | 18.13±0.13 | 1.45±0.47 | **0.77±0.16** | **1.65±0.47** |
| | DeAR [27] | 25.80±0.14 | 18.41±0.12 | 8.09±0.97 | 2.62±0.31 | 8.51±1.00 |
| | VDD [35] | 25.01±0.13 | 18.03±0.13 | 1.70±0.50 | 1.15±0.19 | 2.04±0.48 |
| | **ALA-BA** | 25.57±0.13 | 18.40±0.13 | 1.86±0.53 | 1.37±0.22 | 2.30±0.51 |
| | **ALA-N** | 25.56±0.13 | 18.42±0.13 | **1.39±0.47** | 0.91±0.18 | 1.69±0.43 |

Table 3: Experimental results for VQA open-ended question for bias misalignment on FACET dataset. **Bold** indicates the best result for each baseline, while underline denotes the second-best result.

| VQA-Bias-1 | LLaVA-1.5 | | | PaliGemma | | |
|---|---|---|---|---|---|---|
| | $\|MR_{\mathcal{M}} - MR_{\mathcal{F}}\|$ | $MR_{\mathcal{O}}$ | $MR_{\mathcal{C}}$ | $\|MR_{\mathcal{M}} - MR_{\mathcal{F}}\|$ | $MR_{\mathcal{O}}$ | $MR_{\mathcal{C}}$ |
| Baseline | 3.07±1.18 | 6.14±0.48 | 6.91±0.75 | 3.51±1.07 | 4.44±0.41 | 5.72±0.84 |
| CLIP-clip | 3.82±1.29 | 6.33±0.47 | 7.48±0.84 | 2.12±0.81 | **2.93±0.66** | **1.98±0.27** |
| SFID | 2.97±1.18 | 6.10±0.44 | 6.89±0.70 | **1.03±0.92** | 4.45±0.39 | 4.61±0.45 |
| DeAR | 6.17±1.29 | 6.19±0.46 | 8.76±1.04 | 3.53±1.13 | 4.60±0.38 | 5.86±0.85 |
| VDD | 2.02±1.11 | **5.73±0.47** | 6.09±0.61 | 2.29±1.02 | 4.69±0.42 | 5.25±0.63 |
| **ALA-BA** | 2.86±2.74 | 6.03±1.33 | 6.71±1.86 | 2.55±1.03 | 4.50±0.42 | 5.24±0.73 |
| **ALA-N** | **1.25±0.93** | 5.78±0.45 | **5.96±0.50** | 1.06±0.72 | 3.31±0.34 | 3.50±0.42 |

reducing its dimensionality. However, this direct feature removal is incompatible with encoder-decoder architectures, as it alters the expected representation size. To address this issue, we adapt CLIP-clip for image-to-text tasks using a zero-pruning strategy, which preserves the dimensionality while removing the biased components. In contrast, VDD [35] was originally designed to mitigate hallucination by adjusting the output logits through subtraction of a reference logit derived from an empty or meaningless image. We implement VDD and include it for all evaluation scenarios.

In the SocialCounterfactuals dataset for VQA-Bias-2, intersectional bias arises from a combination of three categories: physical appearance, race, and gender. While comparable debiasing methods can address specific types of bias, CLIP-clip and SFID are primarily effective in mitigating bias within a single category. However, when multiple attributes interact to create intersectional bias in the test set, only DeAR is capable of addressing it. To evaluate their debiasing performance, we report results where CLIP-clip and SFID are applied separately to mitigate bias in race and gender, the only attributes included in the FairFace debiasing training set, as shown in Table 4. In contrast, our method explicitly addresses this issue across different bias types by setting the target bias in stereotypical bias as $s = -1$, non-toxicity, as described in Table 1.

On the other hand, model steering [25] is not included in comparison as it requires computing the gradient of the LMM *w.r.t* the input image, which exceeds our available computational resources.

## 5  Result Analysis

Tables 2, 3, 4, and 5 demonstrate the effectiveness of the proposed method, ALA-BA (Bias Alignment) and ALA-N (Neutralization). Specifically, ALA achieves the best or second-best fairness while minimizing accuracy loss, highlighting the minimal trade-off between utility and fairness. In image captioning (Table 2), ALA demonstrates strong fairness while maintaining caption quality. In the

Table 4: Experimental results for VQA open-ended question for stereotypical bias on SocialCounter-factuals dataset. **Bold** indicates the best result for each baseline, while underline denotes the second and third-best result.

| VQA-Bias-2 | LLaVA-1.5 | | | PaliGemma | | |
|---|---|---|---|---|---|---|
| | $D^P_{\max}$ ($\downarrow$) | $D^R_{\max}$ ($\downarrow$) | $D^G_{\max}$ ($\downarrow$) | $D^P_{\max}$ ($\downarrow$) | $D^R_{\max}$ ($\downarrow$) | $D^G_{\max}$ ($\downarrow$) |
| Baseline | 1.07±0.18 | 0.64±0.17 | 0.40±0.13 | 8.62±1.32 | 6.11±1.37 | 3.52±1.16 |
| CLIP-clip (G) | 2.60±0.48 | 1.78±0.41 | 0.91±0.38 | 7.19±1.10 | 10.94±1.30 | 5.47±1.02 |
| CLIP-clip (R) | 1.50±0.18 | **0.41±0.13** | **0.19±0.11** | **4.46±1.19** | 6.29±1.31 | 2.72±1.09 |
| SFID (G) | 1.09±0.18 | 0.60±0.18 | 0.42±0.14 | 8.07±1.28 | 7.77±1.43 | **1.37±1.04** |
| SFID (R) | 1.08±0.18 | 0.61±0.18 | 0.42±0.14 | 8.17±1.26 | 7.26±1.47 | 1.94±1.09 |
| DeAR | 1.33±0.19 | 0.59±0.16 | 0.36±0.13 | 7.98±1.30 | 5.59±1.29 | 3.52±1.15 |
| VDD | 5.34±0.64 | 1.52±0.49 | 0.58±0.38 | 7.87±1.21 | 6.19±1.29 | **1.02±0.75** |
| **ALA-BA** | 1.04±0.17 | 0.59±0.16 | 0.33±0.14 | 6.50±1.34 | **3.70±1.11** | 3.23±1.19 |
| **ALA-N** | **0.91±0.15** | 0.62±0.16 | 0.27±0.13 | 4.64±0.73 | 4.31±0.77 | 2.49±0.61 |

Table 5: Experimental results for the VQA-as-judge task on the FACET dataset. Red indicates notable degradation. ALA-BA preserves the original model's accuracy, showing no observed degradation, whereas other methods often reduce accuracy level.

| VQA-Bias-3 Accuracy ($\uparrow$) | LLaVA-1.5 | | | PaliGemma | | |
|---|---|---|---|---|---|---|
| | Female | Male | Overall | Female | Male | Overall |
| Baseline | 88.76±0.48 | 86.34±0.32 | 86.96±0.28 | 82.07±0.62 | 86.45±0.33 | 85.32±0.28 |
| CLIP-clip | 89.07±0.50 | 85.97±0.32 | 86.77±0.28 | 79.47±0.63 | 88.22±0.31 | 85.96±0.27 |
| SFID-BA | 88.70±0.49 | 86.34±0.31 | 86.95±0.25 | 82.60±0.59 | 85.83±0.34 | 85.00±0.28 |
| DeAR | 86.53±0.54 | 87.98±0.30 | 87.60±0.26 | 81.60±0.59 | 86.68±0.33 | 85.36±0.28 |
| VDD | 88.38±0.49 | 87.01±0.31 | 87.36±0.26 | 81.61±0.64 | 87.01±0.32 | 85.61±0.30 |
| **ALA-BA** | 88.72±0.48 | 86.34±0.32 | 86.97±0.26 | 82.07±0.58 | 86.41±0.32 | 85.31±0.28 |

VQA open-ended question tasks (Tables 3, 4), ALA consistently achieves top fairness results while preserving accuracy in the VQA-as-judge task (Table 5), whereas representation-based debiasing approaches often degrade utility.

In ALA, the strength of logit adjustment is controlled by the hyperparameter $\lambda$. The ablation study in Appendix C shows that even a small adjustment (e.g., $\lambda = 0.1$) improves fairness, while $\lambda = 2$ provides the best trade-off between utility and fairness. However, excessively large values of $\lambda$ can degrade both performance and fairness, as shown in Figure 4 in Appendix C.

As a limitation of our work, ALA requires external image and text classifiers, resulting in a slight increase in GPU resource usage. However, ALA incurs only a 3.1% increase in GPU utilization, and a 1.2% increase in inference time. These overheads are comparable to those of CLIP-clip, SFID, and DeAR, while ALA remains approximately twice as fast as VDD, which exhibits notably higher inference time. A more detailed analysis of computational costs is provided in Appendix D.

# 6   Conclusion

We introduce Adaptive Logit Adjustment (ALA), a post-hoc debiasing method that refines token probabilities during autoregressive text generation. Unlike existing approaches that modify encoder or decoder representations, ALA directly adjusts logits, mitigating biases without distorting essential model outputs. ALA leverages external classifiers to detect bias misalignment between images and text. It applies gradient-based importance analysis to identify biased tokens and dynamically adjusts token probabilities to align the attributes in input image and generated text. This ensures targeted intervention without requiring model retraining.

Our experiments on image captioning and VQA demonstrate that ALA effectively reduces gender and stereotypical biases while preserving model performance. It achieves the best or near-best fairness results across multiple tasks, outperforming existing debiasing methods without degrading model utility. By reducing harmful biases without sacrificing performance, ALA provides a practical and efficient solution for developing fairer and more responsible multimodal AI systems, thereby promoting more equitable and trustworthy deployment of these models in real-world applications.

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

## A  Bias Alignment with SFID [16]

Selective Feature Imputation for Debiasing (SFID) [16] is designed to obscure bias-related information in the representation space. Specifically, it determines feature importance using a Random Forest classifier [4] trained to predict sensitive attributes. It then imputes values in the most important features with those of the mean of low-confidence samples from the validation dataset, ensuring that all features resemble ambiguous (low-confidence) samples.

However, this method can be applied in a different direction. Instead of obscuring important features, they can be reinforced for certain demographic groups when a clear attribute signal is present, by leveraging high-confidence samples. We adopt this strategy for the VQA-Bias-3 task and report the results of SFID-BA (Bias Alignment) in Table 5.

## B  Evaluation Metric for Image Captioning

METEOR [2] evaluates the trade-off between precision and recall of n-grams in generated captions while accounting for synonym matches. Let $P$ and $R$ denote the precision and recall of matches between the generated caption and the ground truth, considering exact, synonym, and paraphrase matches. METEOR is computed as:

$$\text{METEOR} = F_{\text{mean}} \cdot (1 - \text{Pen})$$

where

$$F_{\text{mean}} = \frac{10 \cdot P \cdot R}{R + 9 \cdot P}$$

represents a harmonic mean, and the penalty term is defined as:

$$\text{Pen} = 0.5 \times \left( \frac{\text{number of chunks}}{\text{number of matches}} \right)^3$$

A chunk refers to a sequence of consecutive words in the generated caption that appear in the reference.

SPICE [1], on the other hand, assesses the semantic quality of captions by comparing sets of propositional semantic tuples extracted from both the candidate and reference captions. It is computed as the F1 score of precision and recall between these tuples, providing a measure of semantic alignment.

## C  Ablation Study

In ALA, the strength of logit adjustment is controlled by the hyperparameter $\lambda$. To analyze its impact, we conduct ablation studies by varying $\lambda$ and evaluating its effect on both performance and fairness in image-to-text tasks.

For VLMs, we assess the effect of $\lambda$ using CLIP-CAP for both **Bias Alignment** and **Neutralization**, as shown in Figure 4 (a). The results indicate that while excessively large $\lambda$ can degrade both performance and fairness, an appropriately chosen $\lambda$, such as $\lambda = 2$, improves fairness without sacrificing performance. Notably, even a small adjustment, such as $\lambda = 0.1$, already leads to noticeable fairness improvements compared to the baseline. This demonstrates that ALA can effectively mitigate bias with minimal intervention, making it adaptable to scenarios with strict performance constraints.

For LMMs, we conduct a similar ablation study using the VQA task on the FACET dataset with LLaVA. Figure 4 (b) illustrates how the fairness metric $MR_C$ for the open-ended description task, VQA-Bias-1, varies with different values of $\lambda$ for each model. Utility is measured separately using a different task, VQA-Bias-3. Similar to the image captioning results in VLMs, fairness improves with moderate values of $\lambda$, such as 2, while excessively large values degrade both fairness and utility.

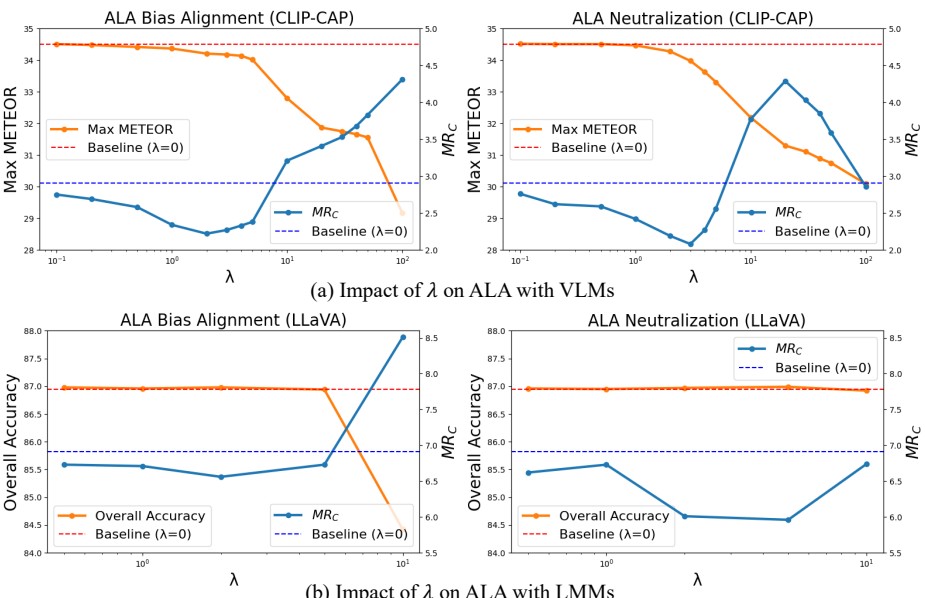

(a) Impact of $\lambda$ on ALA with VLMs

(b) Impact of $\lambda$ on ALA with LMMs

Figure 4: Impact of logit adjustment strength ($\lambda$) on VLMs for image captioning (CLIP-CAP) and LMMs for VQA tasks (LLaVA). The orange curves represent model performance (higher is better): MaxMETEOR score for image captioning and overall accuracy for VQA-as-judge. The blue curves denote fairness, $MR_C$ (lower is better). Moderate values of $\lambda$, such as $\lambda = 2$, improve fairness without degrading performance. Both Bias Alignment (left) and Neutralization (right) exhibit a similar trend, though Neutralization achieves slightly better fairness.

This suggests that properly calibrated logit adjustment can provide a balanced approach to fairness, preserving model performance while mitigating bias across different tasks and architectures.

## D Computational Cost Analysis

As we adopt external image and text classifiers, we carefully examine the additional computational cost. Table 6 shows only a slight increase in RAM and GPU usage, as the external classifiers remain lightweight—a single-layer classifier for image inputs and a two-block transformer for text inputs. Notably, the increases are comparable across all comparison methods. However, VDD exhibits a substantially slower inference time, with a 101.5% increase, as it requires performing inference twice for each input, while our method incurs only a 1.2% increase.

Table 6: Resource consumption comparison of different methods.

| Method | CPU Memory (MB) | | RAM Usage (MB) | | GPU Memory (MB) | | Inference Time (s) | |
|---|---|---|---|---|---|---|---|---|
| | Value | % | Value | % | Value | % | Value | % |
| Baseline | 1368.48 | - | 69578.89 | 0.0 | 13481.79 | 0.0 | 1.5621 | - |
| CLIP-clip | 1630.69 | 19.2 | 69821.79 | 0.3 | 13873.67 | 2.9 | 1.5639 | 0.1 |
| SFID | 1634.55 | 19.4 | 69755.95 | 0.3 | 13873.67 | 2.9 | 1.5739 | 0.8 |
| DeAR | 1406.82 | 2.8 | 69593.04 | 0.0 | 13882.86 | 3.0 | 1.5767 | 0.9 |
| VDD | 1426.94 | 4.3 | 70022.26 | 0.6 | 13876.67 | 2.9 | 3.1472 | 101.5 |
| **Ours (ALA)** | 1615.74 | 18.1 | 70137.92 | 0.8 | 13894.22 | 3.1 | 1.5815 | 1.2 |

## E  Computational Resource

Table 7: Compute Resources Used for Experiments

| Component | Details |
|-----------|---------|
| CPU | AMD EPYC 7313 16-Core Processor |
| GPU | NVIDIA RTX A5000 |

## F  Licenses for existing assets

Table 8: Licenses for each asset

| Dataset | License |
|---------|---------|
| COCO Dataset | CC BY 4.0 |
| FACET Dataset | Research-only |
| SocialCounterfactuals Dataset | MIT License |
| FairFace Dataset | CC BY 4.0 |
| Bias-in-Bios Dataset | MIT License |
| Wikipedia Toxicity Dataset | CC0 License |
| CLIP-CAP | MIT License |
| BLIP | BSD 3-Clause License |
| LLaVA | Apache 2.0 License |
| PaliGemma | Gemma License |

