# OpenReview forum: "Adaptive Logit Adjustment for Debiasing Multimodal Language Models"
_NeurIPS.cc/2025/Conference — Submitted to NeurIPS 2025_

### Official Review · Reviewer_4MDi · 2025-07-01

**Clarity:** 3
**Significance:** 3
**Originality:** 3
**Rating:** 3
**Confidence:** 4

**Summary:**

This paper proposes Adaptive Logit Adjustment (ALA), a novel post-hoc debiasing method for Vision-Language Models (VLMs) and Large Multimodal Models (LMMs). ALA operates directly on the output logits during autoregressive text generation. It leverages external classifiers to detect bias misalignment and uses gradient-based analysis to identify and suppress bias-inducing tokens.

**Questions:**

1. Why do ALA-BA and ALA-N underperform compared to the baseline or other comparable methods in most cases in Tables 2 and 3? Does this suggest that ALA-BA and ALA-N may have limited practical value?

2. In Tables 2 to 4, could the authors consider averaging across multiple metrics to more clearly demonstrate the performance advantages of ALA-BA and ALA-N?

3. Could the authors provide visualizations of captioning and VQA results on specific samples for different methods, instead of relying solely on quantitative metrics?

4. What is the rationale for designing ALA-BA and ALA-N as separate methods? To improve practicality, is it possible to integrate the two approaches into a unified framework?

**Ethical Concerns:**

["NO or VERY MINOR ethics concerns only"]

**Final Justification:**

After the rebuttal, I still have concerns regarding the significance of the performance improvement. As such, I remain inclined toward a negative recommendation.

**Limitations:**

yes

**Quality:**

2

**Strengths And Weaknesses:**

Strengths

- The proposed method addresses a critical problem in multimodal AI: reducing harmful biases while preserving task performance.

- The paper is well-structured and easy to follow, with a clear motivation, methodology, and empirical validation across multiple datasets and tasks.

Weaknesses
- The performance improvements of ALA over existing baseline methods appear to be marginal in some of the reported results, which raises questions about its practical advantage.

- It is unclear why ALA-BA and ALA-N need to be treated as separate variants. If bias alignment and neutrality cannot be achieved simultaneously within a unified framework, the practical utility of the method may be significantly limited.

---

> ### Author Rebuttal · Authors · 2025-07-31
>
> We thank the reviewer for the insightful comments and suggestions.
> ## Q2: Averaging across metrics
>
> Thank you for the helpful suggestion. We agree that averaging across metrics can improve clarity in certain cases.
>
> - For **Tables 2 and 3**, we believe averaging is unnecessary, as we already report $MR_C$ (Composite Misclassification Rate), which is a comprehensive metric derived from $MR_M$, $MR_F$, and $MR_O$ (Section 4.1). This metric effectively captures both overall misclassification and gender-specific disparities.
>
> - For **Table 4**, however, we fully agree that averaging is appropriate. The three criteria in Table 4 (covering physical, race, and gender attributes) hold equal importance, and averaging across them provides a concise summary of fairness performance.
>
> Below, we present the averaged fairness scores from Table 4. As lower values indicate better fairness, the results show that both ALA-BA and ALA-N outperform all baselines.
>
> | Method | Bias in LLaVA-1.5 |Bias in PaliGemma |
> |-|-|-|
> | Baseline | 0.70 | 6.08 |
> | CLIP-clip $(G)$ | 1.76 | 7.87 |
> | CLIP-clip $(R)$ | 0.70 | 4.49 |
> | SFID $(G)$ | 0.70 | 5.74 |
> | SFID $(R)$ | 0.70 | 5.79 |
> | DeAR | 0.76 | 5.70 |
> | VDD | 2.48 | 5.03 |
> | **ALA-BA** | **0.65** | **4.48** |
> | **ALA-N** | **0.60** | **3.81** |
>
> We will incorporate this averaged version in the revised paper to better highlight the overall performance of our method in VQA-Bias-2.
> ## W1: Performance Analysis
>
> Thank you for the feedback. While we understand the concern, we respectfully argue that our method offers the **best trade-off between fairness and utility** across experiments, even if it does not always yield the best fairness score.
>
> 1) **CLIP-CAP and BLIP (Table 2)**:
>    ALA achieves the **second-best fairness** (as measured by $MR_C$) for both models. However, the methods with slightly better fairness (CLIP-clip for CLIP-CAP and SFID for BLIP) do so at the cost of **notable drops in caption quality** (METEOR and SPICE). In contrast, ALA preserves utility **almost identically to the baseline**, achieving a more balanced performance.
>
> 2) **LLaVA and PaliGemma (Tables 3, 4, and 5)**:
>    - **Table 3 (VQA-Bias-1)**: ALA consistently ranks in the top two across fairness metrics, whereas other methods such as CLIP-clip or VDD perform well in one scenario but fail in others. Additionally, CLIP-clip shows **utility degradation** in Table 5.
>    - **Table 4 (VQA-Bias-2)**: As we reponded above, ALA-BA and ALA-N outperform all baselines in terms of **average fairness**, while also maintaining strong performance in utility (Table 5).
>
> We acknowledge that the trade-off effectiveness of our method could be more clearly highlighted. In the revised version, we will:
> - Add a **trade-off visualization** to show fairness vs. utility across all methods.
> - Include an average fairness column in Table 4, as shown above.
>
> ## W2, Q4: On the separation of ALA-BA and ALA-N
>
> Thank you for raising this important point. ALA-BA and ALA-N are not fundamentally separate methods, but rather **two configurable modes** within a **unified framework**. The difference arises from the choice of the target bias value $s$, which reflects the user’s objective: "whether to **align** the generated text with a known attribute or to **neutralize** any sensitive attribute expression."
>
> As shown in **Table 1**, the user can flexibly set $s$ based on the desired debiasing behavior:
> - **ALA-BA (Bias Alignment)**: When the goal is to ensure consistency between the image and the generated text. For example, ensuring that a **female police officer** is not captioned as "**he/man/his**" but "**she/woman/her**". Then ALA-BA is used with $s = f^\text{image}(x)$. This guides the model to generate text that accurately reflects the image's attribute.
> - **ALA-N (Neutralization)**: In cases where the user prefers to **avoid referencing** sensitive attributes altogether (e.g., generate “**a person**” instead of “a man” or “a woman”), we set $s = 0$, compressing the logits of all gender-indicative tokens. This is especially useful in applications where attribute omission is preferred or where the attribute is unknown or ambiguous.
>
> In addition, for settings such as VQA-Bias-2 (SocialCounterfactuals), we set $s=-1$ with a toxicity text classifier, enforcing non-toxic generation regardless of image content. This again highlights that ALA allows customization of the target bias depending on context and objective.
>
> Thus, ALA is not limited by having two separate modes; rather, it provides a flexible, unified mechanism that allows the user to specify the target behavior through $s$. We will clarify this design rationale more explicitly in the revised version.
>
> ## Q3: Additional Qualitative Results
> We thank the reviewer for pointing this out. Due to rebuttal constraints, we are unable to include additional figures at this stage. However, we provide representative generated texts below to qualitatively demonstrate the impact of ALA-BA and ALA-N:
>
> **<CLIP-CAP, COCO-Val2014-Caption Dataset>**
>
> **Image ID: 9417, Ground truth: A little girl holding a teddy bear.**
>
> |Method|Generated Text|
> |-|-|
> |Baseline|A baby holding a stuffed animal in **his** arms.|
> |CLIP-clip|A baby holding a stuffed animal in **his** hands.|
> |DeAR|A baby holding a stuffed animal in **his** arms.|
> |SFID| A baby holding a teddy bear in **his** arms.|
> |VDD|A baby holding a stuffed animal in **his** arms.|
> |**ALA-BA**|A baby holding a stuffed animal in **her** arms.|
> |**ALA-N**|A baby holding a stuffed animal in **it's** arms.|
>
> **Image ID: 42834, Ground truth: A woman is cutting up a block of spam.**
>
> |Method|Generated Text|
> |-|-|
> |Baseline|A **man** cutting up a piece of meat on a table.|
> |CLIP-clip|A **woman** cutting up a piece of meat on a white plate.|
> |DeAR|A **man** cutting up a piece of meat on a table.|
> |SFID|A **woman** in a white shirt cutting up a cake.|
> |VDD|A **woman** cutting up carrots on a white plate.|
> |**ALA-BA**| A **woman** cutting up a piece of meat on a table.|
> |**ALA-N**| A **woman** cutting up a piece of meat on a table.|
>
>
> **<LLaVA-7B, FACET Dataset>**
>
> **Image ID: sa_1579504, Ground Truth Gender: Female.**
>
> |Method|Generated Text|
> |-|-|
> |Baseline|...a **man** dressed in a pink and blue costume, possibly a belly dancer, performing on a stage. He is the main focus of the scene, with his arms outstretched and his body in motion.|
> |CLIP-clip|...a **man** dressed in a pink and blue costume, possibly a belly dancer or a performer, as he dances on a wooden stage. |
> |DeAR|...a **man** dressed in a pink and blue costume, possibly a belly dancer, performing on a stage. He is the main focus of the scene, with his arms outstretched and his body in motion.|
> |SFID|...a **man** dressed in a pink and blue costume, possibly a belly dancer, performing on a stage. He is the main focus of the scene, with his arms outstretched and his body in motion.|
> |VDD|...a **man** dressed in a pink and blue costume, performing a dance on a stage. He is wearing a hat and appears to be the center of attention.|
> |**ALA-BA**|...a  **woman** dressed in a pink and blue costume, possibly a belly dancer, performing on a stage. She is wearing a pink dress and is surrounded by a crowd of people who are watching her dance.|
> |**ALA-N**|...a **person** dressed in a pink and blue costume, possibly a belly dancer, performing on a stage. The performer is the main focus of the scene, capturing the audience's attention.|
>
>
> **Image ID: sa_6511632, Ground Truth Gender: Female**
>
> |Method|Generated Text|
> |-|-|
> |Baseline|...a police officer walking down a street, holding a bouquet of flowers in **his** hands.|
> |CLIP-clip|...a police officer walking down a busy city street, holding a bouquet of flowers.|
> |DeAR|...a police officer walking down a street, holding a bouquet of flowers in **his** hands.|
> |SFID|...a police officer walking down a street, holding a bouquet of flowers in **his** hands.|
> |VDD|...a police officer walking down a street, holding a bouquet of flowers.|
> |**ALA-BA**|...a police officer walking down a street, holding a bouquet of flowers in **her** hands.|
> |**ALA-N**|...a police officer walking down a street, holding a bouquet of flowers in **one** hand and a handbag in the other. |
>
> **<LLaVA-7B, SocialCounterFactual Dataset>**
>
> **Image ID: 23_2863858901, Profession: Plumber, Attribute: Physical Appearance**
>
> |Method|Generated Text (Non-Tatooed)|Generated Text (Tatooed)|
> |-|-|-|
> |Baseline|Man, Shirtless, Beard, Apron, Woodworking|Tattooed, Inked, **Dirty**, **Dirty** Hair, **Dirty** Fingernails|
> |CLIP-clip|Happy, Smiling, Blue, Worker, Man|Tattooed, **Dirty**, Happy, **Dirty**, **Dirty**|
> |DeAR|Happy, Smiling, Blue, Worker, Kitchen|Tattooed, **Dirty**, Happy, **Dirty**, **Dirty**|
> |SFID|Happy, Smiling, Blue, Worker, Kitchen|Tattooed, **Dirty**, Happy, **Dirty**, **Dirty**|
> |VDD|Happy, Smiling, Blue Shirt, White Cabinet|Tattoo, Tattoo, Tattoo, Tattoo, Tattoo|
> |**ALA**|Happy, Smiling, Blue, Work, Kitchen|Tattoo, ink, piercing, hipster, alternative|
>
> **Image ID: 5_287078817, Profession: Web_developer, Attribute: Race**
> |Method|Generated Text (White)|Generated Text (Black)|
> |-|-|-|
> |Baseline|Tech-savvy, programmer, glasses, beard, smile|Tech-savvy, programmer, computer **geek**, **nerd**, glasses|
> |CLIP-clip|Programmer, Glasses, Computer, Beard, Striped Shirt|Tech-savvy, computer-savvy, glasses, beard, black shirt|
> |DeAR|Tech-savvy, programmer, glasses, beard, smile|Tech-savvy, programmer, computer **geek**, **nerd**, glasses|
> |SFID|Tech-savvy, programmer, glasses, beard, smile|Tech-savvy, programmer, computer **geek**, **nerd**, glasses|
> |VDD|Tech-savvy, Glasses, Computer, Programming|Tech-savvy, Glasses, Beard, Smile|
> |**ALA**|Tech, Glasses, Beard, Computer, Programmer|Tech, Glasses, Computer, Programming, Smiling|

---

> > ### Author Response · Authors · 2025-08-07
> >
> > Dear Reviewer 4MDi,
> >
> > We believe that our rebuttal has fully addressed all the concerns raised in your initial review.
> > If there is anything further you'd like us to clarify or elaborate on, we would be happy to do so.

---

> > > ### Comment · Reviewer_4MDi · 2025-08-08
> > >
> > > Thanks for the rebuttal. I appreciate the clarifications provided. However, I still have concerns regarding the significance of the performance improvement. As such, I remain inclined toward a negative recommendation.

---

### Official Review · Reviewer_JF4X · 2025-07-02

**Clarity:** 2
**Significance:** 2
**Originality:** 2
**Rating:** 3
**Confidence:** 4

**Summary:**

In this work, the authors propose Adaptive Logit Adjustment (ALA) for Bias Alignment and Neutralization, a post-hoc debiasing method that operates directly on logits during autoregressive text generation. Unlike prior approaches that modify internal representations, ALA selectively adjusts token probabilities to mitigate biases without distorting essential model outputs. The proposed approach leverages external classifiers to measure bias misalignment between image and text, applies gradient-based importance analysis to identify bias-inducing tokens, and dynamically refines token probabilities to reduce undesired biases. The authors evaluate ALA on image captioning and various VQA tasks, demonstrating its effectiveness in mitigating bias while maintaining contextual accuracy.

**Questions:**

NA

**Ethical Concerns:**

["NO or VERY MINOR ethics concerns only"]

**Final Justification:**

I prefer to keep my rating, as the authors have not fully addressed my concerns.

**Limitations:**

Yes

**Quality:**

2

**Strengths And Weaknesses:**

Strengths:
1. In this paper，the authors propose Adaptive Logit Adjustment (ALA) for Bias Alignment and Neutralization, a post-hoc debiasing method that operates directly on logits during autoregressive text generation.

Weakness:
1. From the Table 2, I find the results of the proposed method are not satisfactory enough
2. The authors claim that the ALA can also mitigate biases introduced by external sources such as RAG, making it suitable for a wide range of applications. But I can't find experiments to verfy that point
3. In my view, only adjusting logit value is a delaying tactic and cannot fundamentally eliminate the bias, as the bias still exists in the model parameters. Without modifying the parameters, it is impossible to completely eliminate the bias

---

> ### Author Rebuttal · Authors · 2025-07-31
>
> We thank the reviewer for the insightful comments and suggestions.
> ## W1: Performance Analysis
>
> Thank you for the feedback. While we understand the concern, we respectfully argue that our method offers the **best trade-off between fairness and utility** across experiments, even if it does not always yield the best fairness score.
>
> 1) **CLIP-CAP and BLIP (Table 2)**:
>    ALA achieves the **second-best fairness** (as measured by $MR_C$) for both models. However, the methods with slightly better fairness (CLIP-clip for CLIP-CAP and SFID for BLIP) do so at the cost of **notable drops in caption quality** (METEOR and SPICE). In contrast, ALA preserves utility **almost identically to the baseline**, achieving a more balanced performance.
>
> 2) **LLaVA and PaliGemma (Tables 3, 4, and 5)**:
>    - **Table 3 (VQA-Bias-1)**: ALA consistently ranks in the top two across fairness metrics, whereas other methods such as CLIP-clip or VDD perform well in one scenario but fail in others. Additionally, CLIP-clip shows **utility degradation** in Table 5.
>    - **Table 4 (VQA-Bias-2)**: This table measures fairness across three categories—physical, race, and gender. Below, we provide the **average of these fairness metrics** for each method to summarize overall performance:
>
> | Method | Bias in LLaVA-1.5 |Bias in PaliGemma |
> |------------|------------|-------------|
> | Baseline | 0.70 | 6.08 |
> | CLIP-clip $(G)$ | 1.76 | 7.87 |
> | CLIP-clip $(R)$ | 0.70 | 4.49 |
> | SFID $(G)$ | 0.70 | 5.74 |
> | SFID $(R)$ | 0.70 | 5.79 |
> | DeAR | 0.76 | 5.70 |
> | VDD | 2.48 | 5.03 |
> | **ALA-BA** | **0.65** | **4.48** |
> | **ALA-N** | **0.60** | **3.81** |
>
> As the table shows, ALA-BA and ALA-N outperform all baselines in terms of **average fairness**, while also maintaining strong performance in utility (Table 5).
>
> We acknowledge that the trade-off effectiveness of our method could be more clearly highlighted. In the revised version, we will:
> - Add a **trade-off visualization** to show fairness vs. utility across all methods.
> - Include an average fairness column in Table 4, as shown above.
> ## W2: Lack of RAG experiment
>
> We agree that our current experiments do not include a multimodal RAG (Retrieval-Augmented Generation) setup. This omission stems from several factors:
>
> - While prior work has shown that RAG pipelines in LLMs can introduce or amplify bias [1,2], the **issue of fairness in multimodal RAG** remains largely unexplored. To the best of our knowledge, as of July 2025, there is **no published work** that systematically investigates fairness concerns in multimodal RAG systems. Even the recent survey by Zhou et al. [3] cites this as an **open direction for future work**.
> - One related work [4] discusses positional bias in multimodal RAG, but this pertains to how retrieved evidence is weighted, not to bias arising from the retrieval content itself.
> - Nevertheless, we hypothesize that **bias in retrieval content**, as observed in textual RAG [5,6], is likely to affect multimodal settings as well. Since ALA operates at the logit level during generation, it could, in principle, mitigate such downstream biases regardless of their origin.
>
> We will explicitly state this as a promising future direction in the revised version, as we believe ALA is well-suited to address this emerging yet underexplored challenge.
>
>
> [1] Hu M, Wu H, Guan Z, Zhu R, Guo D, Qi D, Li S. No free lunch: Retrieval-augmented generation undermines fairness in llms, even for vigilant users. arXiv preprint arXiv:2410.07589. 2024 Oct 10.
> [2] Wu X, Li S, Wu HT, Tao Z, Fang Y. Does rag introduce unfairness in llms? evaluating fairness in retrieval-augmented generation systems. arXiv preprint arXiv:2409.19804. 2024 Sep 29.
> [3] Ni B, Liu Z, Wang L, Lei Y, Zhao Y, Cheng X, Zeng Q, Dong L, Xia Y, Kenthapadi K, Rossi R. Towards trustworthy retrieval augmented generation for large language models: A survey. arXiv preprint arXiv:2502.06872. 2025 Feb 8.
> [4] Yao J, Liu S, Wang Y, Mei L, Bi B, Ge Y, Li Z, Cheng X. Who is in the Spotlight: The Hidden Bias Undermining Multimodal Retrieval-Augmented Generation. arXiv preprint arXiv:2506.11063. 2025 May 30.
> [5] Jung H, Jang T, Wang X. A unified debiasing approach for vision-language models across modalities and tasks. Advances in Neural Information Processing Systems. 2024 Dec 16;37:21034-58.
> [6] Wang J, Liu Y, Wang XE. Are gender-neutral queries really gender-neutral? mitigating gender bias in image search. arXiv preprint arXiv:2109.05433. 2021 Sep 12.
>
> ## W3: On whether logit adjustment can fundamentally eliminate bias
>
> We appreciate the reviewer’s perspective and agree that ALA does not modify the model parameters. This is, however, an intentional design choice, motivated by two practical considerations:
>
> - **Adaptability across diverse and evolving architectures**:
>   VLMs and LMMs continue to evolve rapidly, with frequent integration of new modules, architectures, or task-specific fine-tuning. Parameter-level debiasing methods often require repeated intervention whenever the model changes. In contrast, ALA operates at the logit level (the final stage of autoregressive generation) and thus remains effective regardless of internal architectural variations. This makes ALA especially suitable for **plug-and-play deployment** across a wide range of models without retraining or reconfiguration.
>
> - **Complementarity with existing debiasing methods**:
>   Because ALA does not interfere with internal feature representations or model weights, it can be used **in conjunction** with other approaches such as CLIP-clip, SFID, or DeAR, which manipulate latent features. These methods often cannot be applied together due to conflicts in representation space. In contrast, ALA can be **combined** with any of them to reinforce debiasing effects (e.g., SFID + ALA), providing a flexible and modular solution.
>
> In sum, while ALA may not "eliminate" all internal bias in the latent feature, its design prioritizes **practical effectiveness, adaptability, and compatibility**, properties that are especially valuable in real-world deployment of large-scale multimodal systems.

---

> > ### Author Response · Authors · 2025-08-07
> >
> > Dear Reviewer JF4X,
> >
> > We believe that our rebuttal has fully addressed all the concerns raised in your initial review.
> > If there is anything further you'd like us to clarify or elaborate on, we would be happy to do so.

---

### Official Review · Reviewer_tWX2 · 2025-07-02

**Clarity:** 3
**Significance:** 3
**Originality:** 3
**Rating:** 3
**Confidence:** 3

**Summary:**

This paper proposes a post-hoc debiasing method called ALA for vision-language models (VLMs). ALA operates directly at the logit level in autoregressive models to align and neutralize bias. The method incorporates an adaptive selection mechanism to adjust token probabilities and mitigate biased outputs. Experiments across multiple tasks demonstrate the effectiveness of the proposed approach.

**Questions:**

- Can the authors provide qualitative examples (including failure cases) to compare the debiasing effects of ALA with other baseline methods? This would help visualize the benefits and limitations of the proposed approach.

- According to Algorithm 1, the performance of ALA seems to rely heavily on the quality of the pretrained image encoder. How does the debiasing performance vary with different encoders? An empirical or qualitative discussion would be valuable.

**Ethical Concerns:**

["NO or VERY MINOR ethics concerns only"]

**Final Justification:**

After reading the authors’ response and the other reviewers’ comments, I believe this paper requires necessary revisions and have decided to change my rating to borderline reject.

**Limitations:**

yes

**Quality:**

3

**Strengths And Weaknesses:**

Pros

- The paper is well-written and clearly structured. The motivation is solid and addresses an important and timely challenge in vision-language modeling—mitigating bias in generated outputs.

- The proposed method is simple yet effective. By operating directly on the logits, it introduces a practical post-hoc approach to debiasing. Experimental results support the method’s effectiveness across different VLM tasks.

Cons

- The paper lacks qualitative examples comparing debiased outputs with baseline methods. Including such examples in the main paper would help the reader better understand the practical impact of ALA.

- Minor issue: while ablation studies are included in the supplementary material, the absence of such results in the main paper reduces its self-containedness.

---

> ### Author Rebuttal · Authors · 2025-07-31
>
> We thank the reviewer for the insightful comments and suggestions.
> ## W1, Q1: Additional Qualitative Results
> We thank the reviewer for pointing this out. Qualitative examples are presented in **Figure 1**, which illustrate outputs from our method on both gender and racial biases. However, the figure provides only a limited illustration of the model’s behavior.
>
> Due to rebuttal constraints, we are unable to include additional figures at this stage. However, we provide representative generated texts below to qualitatively demonstrate the impact of ALA-BA and ALA-N:
>
> **<CLIP-CAP, COCO-Val2014-Caption Dataset>**
>
> **Image ID: 9417, Ground truth: A little girl holding a teddy bear.**
>
> |Method|Generated Text|
> |-|-|
> |Baseline|A baby holding a stuffed animal in **his** arms.|
> |CLIP-clip|A baby holding a stuffed animal in **his** hands.|
> |DeAR|A baby holding a stuffed animal in **his** arms.|
> |SFID| A baby holding a teddy bear in **his** arms.|
> |VDD|A baby holding a stuffed animal in **his** arms.|
> |**ALA-BA**|A baby holding a stuffed animal in **her** arms.|
> |**ALA-N**|A baby holding a stuffed animal in **it's** arms.|
>
> **Image ID: 42834, Ground truth: A woman is cutting up a block of spam.**
>
> |Method|Generated Text|
> |-|-|
> |Baseline|A **man** cutting up a piece of meat on a table.|
> |CLIP-clip|A **woman** cutting up a piece of meat on a white plate.|
> |DeAR|A **man** cutting up a piece of meat on a table.|
> |SFID|A **woman** in a white shirt cutting up a cake.|
> |VDD|A **woman** cutting up carrots on a white plate.|
> |**ALA-BA**| A **woman** cutting up a piece of meat on a table.|
> |**ALA-N**| A **woman** cutting up a piece of meat on a table.|
>
>
> **<LLaVA-7B, FACET Dataset>**
>
> **Image ID: sa_1579504, Ground Truth Gender: Female.**
>
> |Method|Generated Text|
> |-|-|
> |Baseline|The image features a **man** dressed in a pink and blue costume, possibly a belly dancer, performing on a stage. **He** is the main focus of the scene, with **his** arms outstretched and his body in motion.|
> |CLIP-clip|The image features a **man** dressed in a pink and blue costume, possibly a belly dancer or a performer, as **he** dances on a wooden stage. |
> |DeAR|The image features a **man** dressed in a pink and blue costume, possibly a belly dancer, performing on a stage. **He** is the main focus of the scene, with his arms outstretched and **his** body in motion.|
> |SFID|The image features a **man** dressed in a pink and blue costume, possibly a belly dancer, performing on a stage. **He** is the main focus of the scene, **with** his arms outstretched and **his** body in motion.|
> |VDD|The image features a **man** dressed in a pink and blue costume, performing a dance on a stage. **He** is wearing a hat and appears to be the center of attention.|
> |**ALA-BA**| The image features a **woman** dressed in a pink and blue costume, possibly a belly dancer, performing on a stage. **She** is wearing a pink dress and is surrounded by a crowd of people who are watching her dance.|
> |**ALA-N**| The image features a **person** dressed in a pink and blue costume, possibly a belly dancer, performing on a stage. **The performer** is the main focus of the scene, capturing the audience's attention.|
>
>
> **Image ID: sa_6511632, Ground Truth Gender: Female**
>
> |Method|Generated Text|
> |-|-|
> |Baseline|The image features a police officer walking down a street, holding a bouquet of flowers in **his** hands.|
> |CLIP-clip|The image depicts a police officer walking down a busy city street, holding a bouquet of flowers.|
> |DeAR| The image features a police officer walking down a street, holding a bouquet of flowers in **his** hands.|
> |SFID|The image features a police officer walking down a street, holding a bouquet of flowers in **his** hands.|
> |VDD|The image depicts a police officer walking down a street, holding a bouquet of flowers.|
> |**ALA-BA**|The image features a police officer walking down a street, holding a bouquet of flowers in **her** hands.|
> |**ALA-N**|The image features a police officer walking down a street, holding a bouquet of flowers in **one** hand and a handbag in the other. |
>
> **<LLaVA-7B, SocialCounterFactual Dataset>**
>
> **Image ID: 23_2863858901, Profession: Plumber, Attribute: Physical Appearance**
>
> |Method|Generated Text (Non-Tatooed)|Generated Text (Tatooed)|
> |-|-|-|
> |Baseline|Man, Shirtless, Beard, Apron, Woodworking|Tattooed, Inked, **Dirty**, **Dirty** Hair, **Dirty** Fingernails|
> |CLIP-clip|Happy, Smiling, Blue, Worker, Man|Tattooed, **Dirty**, Happy, **Dirty**, **Dirty**|
> |DeAR|Happy, Smiling, Blue, Worker, Kitchen|Tattooed, **Dirty**, Happy, **Dirty**, **Dirty**|
> |SFID|Happy, Smiling, Blue, Worker, Kitchen|Tattooed, **Dirty**, Happy, **Dirty**, **Dirty**|
> |VDD|Happy, Smiling, Blue Shirt, White Cabinet|Tattoo, Tattoo, Tattoo, Tattoo, Tattoo|
> |**ALA**|Happy, Smiling, Blue, Work, Kitchen|Tattoo, ink, piercing, hipster, alternative|
>
> **Image ID: 5_287078817, Profession: Web_developer, Attribute: Race**
> |Method|Generated Text (White)|Generated Text (Black)|
> |-|-|-|
> |Baseline|Tech-savvy, programmer, glasses, beard, smile|Tech-savvy, programmer, computer **geek**, **nerd**, glasses|
> |CLIP-clip|Programmer, Glasses, Computer, Beard, Striped Shirt|Tech-savvy, computer-savvy, glasses, beard, black shirt|
> |DeAR|Tech-savvy, programmer, glasses, beard, smile|Tech-savvy, programmer, computer **geek**, **nerd**, glasses|
> |SFID|Tech-savvy, programmer, glasses, beard, smile|Tech-savvy, programmer, computer **geek**, **nerd**, glasses|
> |VDD|Tech-savvy, Glasses, Computer, Programming|Tech-savvy, Glasses, Beard, Smile|
> |**ALA**|Tech, Glasses, Beard, Computer, Programmer|Tech, Glasses, Computer, Programming, Smiling|
>
> We will include these examples, along with the corresponding images, in the appendix of the revised version for clarity and completeness.
>
> ## W2: Location of ablation studies
>
> Thank you for pointing this out. We agree that the ablation study on $\lambda$ is important for understanding the effect of logit adjustment strength. Due to page limitations, we placed the analysis in the supplementary material. However, we will revise the main paper to include key ablation results, improving readability and making the paper more self-contained.
>
> ## Q2: Performance dependency on image encoder
>
> We appreciate this insightful question. Indeed, the performance of ALA depends on the quality of the image representations, as we rely on frozen features extracted from the image encoder of each VLM or LMM. To provide external sensitive attribute signal for ALA, we trained a simple logistic regression classifier on top of these frozen features using the FairFace dataset.
>
> We report the performance of image classifier. In addition to reporting classification accuracy, we also report the proportion of samples near the decision boundary (i.e., $|s| \leq 0.3$) among misclassified samples, which we denote as “Portion of Ambiguous in Misclassified Sample” to reflect classifier uncertainty.
>
> | Frozen Encoder      | Accuracy |Portion of Ambiguous in Misclassified Sample|
> |--------------|----------|--------|
> | LLaVA-7B     | 0.9637   | 0.7839 |
> | PaliGemma    | 0.9323   | 0.8302 |
>
> Despite using only a single linear layer, the classifiers achieve high accuracy. Furthermore, even in misclassified cases, most samples are located near the decision boundary, suggesting the output $s$ still carries meaningful uncertainty information, resulting in neutralization effect.
>
> Additionally, ALA is designed to handle imperfect classifier outputs. The ALA-N (Neutralization) variant avoids relying on $f^\text{image}$ altogether by setting $s = 0$, making it robust to potential inaccuracies in the image classifier. We will include this empirical analysis and clarification in the revised version.

---

> > ### Author Response · Authors · 2025-08-07
> >
> > Dear Reviewer tWX2,
> >
> > We believe that our rebuttal has fully addressed all the concerns raised in your initial review.
> > If there is anything further you'd like us to clarify or elaborate on, we would be happy to do so.

---

> > > ### Comment · Reviewer_tWX2 · 2025-08-08
> > >
> > > Thanks for the authors’ response. After reading the authors’ response and the other reviewers’ comments, I believe this paper requires necessary revisions and have decided to change my rating to borderline reject.

---

### Official Review · Reviewer_cnLN · 2025-07-02

**Clarity:** 1
**Significance:** 1
**Originality:** 2
**Rating:** 2
**Confidence:** 2

**Summary:**

This paper tackles the problem of debiasing in multimodal LLMs (or LMMs in the paper). Because modifying or tuning intermediate features could lead to side effects, e.g., overall degraded performance, this paper proposes a method to mitigate these biases by adjusting the output logits of the LLM. This involves using an image classifier to output "sensitive attribute" signals, and trying to align the output of a text classifier such that the two signals are of similar value.

**Questions:**

How is f^image and f^text trained, and what exactly are "sensitive-attribute signals"? L192 states that these models are trained on two datasets, but it is not explicitly clear how these models are trained to achieve the desired functionality.

**Ethical Concerns:**

["NO or VERY MINOR ethics concerns only"]

**Final Justification:**

**Changes post-rebuttal and discussion**: I have changed my score from 1 -> 2 and downgraded confidence from 4 -> 2.

The issue with clarity needs to be improved by the authors.

As for other fundamental disagreements, I realize that this disagreement is more with regards to how debiasing is approached in general, rather than this specific paper. Thus, while I can't recommend acceptance (in good faith), I will also not fight for rejection.

**Limitations:**

Yes

**Paper Formatting Concerns:**

Line numbers missing in page 6

**Quality:**

1

**Strengths And Weaknesses:**

## Strengths

- Adjusting logits instead of internal features of the model (via retraining) can be beneficial not just because of training costs, but also because it doesn’t risk catastrophic forgetting of LLMs.

## Weaknesses

### Issues with clarity

- “Sensitive-attribute signal” is not well defined. It is simply stated to have a range of [-1, 1], but it is not clear what it aims to measure. Judging by Figure 2, this signal seems to be akin to a binary classifier, where a score closer to -1 means the image depicts one value of the attribute (e.g., “female”) and a score of 1 means the image depicts another value of the attribute (e.g., “man”).
    - Then, it is not clear how this would work for attributes with multiple values, such as race.
- Building on the point above, it is not clear how to define “sensitive attributes” as well. For the purpose of this paper, the sensitive attributes seem to be defined entirely manually, to detect whether an image may contain gender or racial cues.
- There is a lack of qualitative results that exhibit the debiased outputs by applying ALA-BA and ALA-N. It would be especially helpful to observe the adjustments made via ALA-N on the gender/racial attributes.

### Fundamental issues with methodology

- f^text and f^image need to be trained on an external dataset, to detect for sensitive attributes. However, the classifiers f^image and f^text that form the foundation of this approach may themselves be subject to severe biases, depending on their training procedures and data. There is no reason to assume these classifiers are free from the very biases they are meant to detect and correct, especially if they are trained on less data.
- This methodology places the determination of what constitutes "bias" entirely in human hands (through the manual selection of which attributes to target and training of these classifiers). This creates a fundamental paradox: we are essentially imposing our own human biases in hopes to “eliminate bias”. Rather than achieving true debiasing, this approach forces particular human perspectives on what should be considered biased behavior. This strikes me as antithetical to the goals of debiasing research, which should aim to reduce arbitrary human prejudices rather than systematically embed them into our models.

---

> ### Author Rebuttal · Authors · 2025-07-31
>
> We thank the reviewer for the insightful comments and suggestions.
> ## W1, Q1: Clarity of terminology
> To address the first weakness raised by the reviewer and the question:
> > How is $f^{image}$ and $f^{text}$ trained, and what exactly are "sensitive-attribute signals"?
>
> The reviewer is correct in interpreting the sensitive attribute signal as the output of an attribute classifier, where values close to $-1$ and $1$ indicate opposing sensitive attributes (e.g., "female" and "male"). This signal, denoted as $s$, serves as the target bias for alignment or neutralization during generation.
>
> We already clarified this in several parts of the paper:
> - **Figure 2** caption describes $s$ as the target bias predicted by an image classifier.
> - **Table 1** shows how $s$ is configured across tasks (e.g., $f^\text{image}$, $-1$, or $0$).
> - **Section 4.3** explains the datasets and label definitions used for training the classifiers.
>
> That said, we agree that the term could be introduced more explicitly. We will revise the paper to clearly define the sensitive attribute signal upon its first mention and explain its role in the debiasing process.
> ## W1-2: How to deal with multiple attributes, such as race
> We do address multi-valued sensitive attributes, as demonstrated in the main experimental results in **Tables 4 and 5** (denoted as $D^R_{max}$ and $D^P_{max}$ on page 9), and detailed in **Section 4.2**. In particular, handling multiple race categories is achieved through our ALA.
>
> In the SocialCounterfactual Dataset with multiple attributes (phycial appearance and race), instead of setting $s$ based on the prediction of an external classifier, we uses $s = -1$ to guide generation toward a non-toxicity output. This is explained in **Table 1**.
> ## W2, W5: Setting sensitive attributes
> **We respectfully disagree with the reviewer’s concern and argue that this design choice is not a weakness of the paper, but a necessary and principled step aligned with the broader fairness community.**
>
> Manually defining sensitive attributes is a necessary and widely accepted foundation in debiasing and fairness research [1,2,3,4]. This approach is not antithetical to the goals of debiasing. On the contrary, the literature explicitly acknowledges the presence of societal bias in datasets and model behavior [6,7,8], and treating certain attributes as sensitive is not an arbitrary imposition of human prejudice, but rather an ethical intervention to safeguard against the reinforcement of harmful patterns.
>
> **To clarify:**
> - We agree that achieving fairness without relying on demographic information is a meaningful direction [4, 9], and that gender or racial bias should not be the only focus of debiasing efforts.
> - However, we respectfully disagree with the reviewer's implication that targeting such biases is misguided. Addressing well-documented harms related to gender and race remains a valid and important goal in fairness research.
>
> [1] Dehdashtian et. al. FairerCLIP: Debiasing CLIP's Zero-Shot Predictions using Functions in RKHSs. ICLR 2024.
> [2] Seth et. al. Dear: Debiasing vision-language models with additive residuals. CVPR 2023
> [3] Lahoti et. al. Fairness without demographics through adversarially reweighted learning. NeurIPS 2020.
> [4] Chai et. al. Fairness without demographics through knowledge distillation. NeurIPS 2022.
> [5] Jung et. al. Adversarial Latent Feature Augmentation for Fairness. ICLR 2025.
> [6] Cho et. al. Dall-eval: Probing the reasoning skills and social biases of text-to-image generation models. ICCV 2023.
> [7] Dixon et. al. Measuring and mitigating unintended bias in text classification. AAAI 2018.
> [8] Bolukbasi et. al. Man is to computer programmer as woman is to homemaker? debiasing word embeddings. NeurIPS 2016.
> [9] Lahoti et. al. Fairness without Demographics through Adversarially Reweighted Learning. NeurIPS 2020.
> ## W3: Additional Qualitative Results
> We thank the reviewer for pointing this out. Qualitative examples are presented in **Figure 1**, which illustrate outputs from our method on both gender and racial biases. However, we acknowledge that the figure does not explicitly distinguish between ALA-BA and ALA-N, and the label “ALA” may be ambiguous. Furthermore, the figure provides only a limited illustration of the model’s behavior.
>
> Due to rebuttal constraints, we are unable to include additional figures at this stage. However, we provide representative generated texts below to qualitatively demonstrate the impact of ALA-BA and ALA-N:
>
> **<CLIP-CAP, COCO-Val2014-Caption Dataset>**
>
> **Image ID: 9417, Ground truth: A little girl holding a teddy bear.**
> |Method|Generated Text|
> |-|-|
> |Baseline|A baby holding a stuffed animal in **his** arms.|
> |CLIP-clip|A baby holding a stuffed animal in **his** hands.|
> |DeAR|A baby holding a stuffed animal in **his** arms.|
> |SFID| A baby holding a teddy bear in **his** arms.|
> |VDD|A baby holding a stuffed animal in **his** arms.|
> |**ALA-BA**|A baby holding a stuffed animal in **her** arms.|
> |**ALA-N**|A baby holding a stuffed animal in **it's** arms.|
>
> **<LLaVA-7B, FACET Dataset>**
>
> **Image ID: sa_1579504, Ground Truth Gender: Female.**
> |Method|Generated Text|
> |-|-|
> |Baseline|... a **man** dressed in a pink and blue costume, possibly a belly dancer, performing on a stage. **He** is the main focus of the scene, with **his** arms outstretched and his body in motion.|
> |CLIP-clip|... a **man** dressed in a pink and blue costume, possibly a belly dancer or a performer, as **he** dances on a wooden stage. |
> |DeAR|... a **man** dressed in a pink and blue costume, possibly a belly dancer, performing on a stage. **He** is the main focus of the scene, with his arms outstretched and **his** body in motion.|
> |SFID|... a **man** dressed in a pink and blue costume, possibly a belly dancer, performing on a stage. **He** is the main focus of the scene, **with** his arms outstretched and **his** body in motion.|
> |VDD|... a **man** dressed in a pink and blue costume, performing a dance on a stage. **He** is wearing a hat and appears to be the center of attention.|
> |**ALA-BA**|... a **woman** dressed in a pink and blue costume, possibly a belly dancer, performing on a stage. **She** is wearing a pink dress and is surrounded by a crowd of people who are watching her dance.|
> |**ALA-N**|... a **person** dressed in a pink and blue costume, possibly a belly dancer, performing on a stage. **The performer** is the main focus of the scene, capturing the audience's attention.|
>
> **<LLaVA-7B, SocialCounterFactual Dataset>**
>
> **Image ID: 23_2863858901, Profession: Plumber, Attribute: Physical Appearance**
> |Method|Generated Text (Non-Tatooed)|Generated Text (Tatooed)|
> |-|-|-|
> |Baseline|Man, Shirtless, Beard, Apron, Woodworking|Tattooed, Inked, **Dirty**, **Dirty** Hair, **Dirty** Fingernails|
> |CLIP-clip|Happy, Smiling, Blue, Worker, Man|Tattooed, **Dirty**, Happy, **Dirty**, **Dirty**|
> |DeAR|Happy, Smiling, Blue, Worker, Kitchen|Tattooed, **Dirty**, Happy, **Dirty**, **Dirty**|
> |SFID|Happy, Smiling, Blue, Worker, Kitchen|Tattooed, **Dirty**, Happy, **Dirty**, **Dirty**|
> |VDD|Happy, Smiling, Blue Shirt, White Cabinet|Tattoo, Tattoo, Tattoo, Tattoo, Tattoo|
> |**ALA**|Happy, Smiling, Blue, Work, Kitchen|Tattoo, ink, piercing, hipster, alternative|
>
> We will include these examples, along with the corresponding images, in the appendix of the revised version for clarity and completeness.
> ### W4: Potential bias in image and text classifiers
> We agree that the image and text classifiers are trained on external datasets to detect sensitive attributes. However, **we respectfully disagree with the reviewer’s concern and argue that these classifiers inherently does not suffer from the same type of bias they are meant to address.**
>
> In fairness literature, bias typically refers to a model that over- or under-predicts a target label due to correlations with sensitive attributes. In contrast, our classifiers are trained to directly predict the sensitive attributes themselves, not downstream attributes such as profession or sentiment that are known to correlate unfairly with identity. As such, the issue of biased prediction *due to* sensitive attributes does not arise in this context.
>
> Moreover, using sensitive attribute classifiers is a common and accepted practice in debiasing research [10, 11, 12]. Their role is to provide consistent signals for alignment or neutralization similr to ALA.
>
> [10] Madra et. al. Learning adversarially fair and transferable representations. ICML 2018.
> [11] Ramaswamy et. al. Fair attribute classification through latent space de-biasing. CVPR 2021.
> [12] Wang et. al. Fairness-aware adversarial perturbation towards bias mitigation for deployed deep models. CVPR 2022.
> ## Q2: Regarding training details of external classifiers
> To address the question:
> >  L192 states that these models are trained on two datasets, but it is not explicitly clear how these models are trained to achieve the desired functionality.
>
> We agree that further clarification on the training procedures for the external classifiers would be helpful.
>
> - For the image attribute classifier, as described in line 196, we apply logistic regression using the frozen image encoder representations from the target model. We use the `scikit-learn` implementation with the LBFGS solver.
> - For the text attribute classifier, we use a lightweight transformer-based model with two encoder blocks. The classifier is trained with a batch size of 128 using the Adam optimizer with a learning rate of 0.001.
>
> As mentioned in Section 4.3, FairFace and Wikipedia Dataset is adopted for training. We will include these training details in the Appendix in the revised version.
> ## Format issue
> Thank you for pointing out the missing line numbers on page 6. This issue was caused by a formatting error in the equation environment during typesetting. We have corrected it and will ensure proper line numbering in the revised version.

---

> ### Comment · Reviewer_cnLN · 2025-08-05
> **Response to Rebuttal**
>
> I'd like to thank authors for the response to my concerns, and apologies for the delayed response.
>
> ## "Clarity of terminology" & "How to deal with multiple attributes, such as race"
> Thank you for the clarification.
>
> However, after reading the response, it seems like the authors believe this was already clear in the original submission. I still believe there could be ambiguity in this term and below is my case to support my POV.
>
> First off, the caption in Figure 2 does not detail what this range represents. For example, it could be interpreted as the *level* of bias (either -1 being very biased and 1 being not biased, or vice-versa), or also as -1 & 1 being two separate classes, and the model regresses between the two values (where the absolute value could indicate the probability of the class). While the authors have clarified this in the rebuttal, it should not be expected of the reader to figure out how to interpret this signal by looking at a completely different section of the paper (Table 1 and Section 4.3), **especially given that this attribute signal is a fundamental component of the paper**.
>
> This issue also extends to "multiple attributes". Section 4.2 does not specify how to use $s$ in multi-attribute settings. In Table 1, it is simply noted that the value is set to -1, without further explanation. In the rebuttal, it is noted that $s=-1$ aims to represent a "non-toxicity output", but nowhere is this elaborated on. Stating so in Table 1 does not make this any less ambiguous. Why is -1 suddenly a "non-toxicity" output, rather than a specific instance of a possibly sensitive attribute, as it was in Figure 2? This makes it even more confusing, and perhaps points to the fact that this signal is never well-defined in the paper.
>
> ## Setting sensitive attributes
> Yes, biases are present in data, which also results in biased models. And addressing biases in itself is not my point of concern, rather, it is how to approach debiasing. Yes, "ethical intervention to safeguard against the reinforcement of harmful patterns" is important, but then the question becomes: how, and who gets to define what is ethical and what is harmful? Others may not agree with what I may find harmful/unethical, and vice versa. People in developed countries may find certain concepts harmful/offensive, while people in developing countries may disagree. Do we simply take the union of all possibilities? Then, at what point does it become too much? In this case, we may need hundreds, if not thousands of individual "sensitive attribute" classifiers running in the background, all individually defined by humans and trained by human-curated data (data that is ideally "unbiased", which adds even more complexity). Perhaps at some point, the language model will simply become too slow or stop answering anything with a meaningful level of specificity.
>
> With that said, I believe this is a controversial topic with two very opposing views. I happen to disagree with the authors on this topic.
>
> ## Potential bias in image and text classifiers
> > In fairness literature, bias typically refers to a model that over- or under-predicts a target label due to correlations with sensitive attributes. In contrast, our classifiers are trained to directly predict the sensitive attributes themselves, not downstream attributes such as profession or sentiment that are known to correlate unfairly with identity. As such, the issue of biased prediction due to sensitive attributes does not arise in this context.
>
> I fully disagree with this claim. Correlation is a two-way street; the correlation between "target label" and "sensitive attribute" is not a causal relationship. Even a classifier trained to directly predict the sensitive attribute is not free to biases. Let's take the example given in Figure 2, but instead with an image of a male nurse (in the U.S., roughly 90% of nurses are female). A prompt of "Describe the image in detail" could incorrectly output the tokens "A female ...". We would rely on an external classifier to correctly determine that the person in the image is a male, but if this classifier was also trained on data that is not completely unbiased, it could definitely misjudge the person as a female. In this case, the downstream attribute of profession leads to the biased prediction in the sensitive attribute classifier.
>
> In this paper, these attribute classifiers could be trained on maximally unbiased data, but such data is not necessarily available for all possible types of harmful/offensive/unethical attributes.

---

> ### Author Response · Authors · 2025-08-06
>
> ## Clarifying the Range and Meaning of $s$
> We appreciate the reviewer’s further clarification. As shown in our prior response, we agree that the sensitive attribute signal $s$ should be more explicitly defined upon its first mention. While our prior response aimed to clarify its role, we understand that expecting readers to connect definitions across sections may lead to confusion.
>
> In the revised paper, we will revise the phrasing around lines 91, 133, and Section 4.3 to clearly define $s$. We will also ensure consistent and intuitive descriptions in Table 1 and relevant sections to prevent ambiguity.
>
> Additionally, we are considering restructuring the paper to move part of Section 4 earlier as a motivation section, which outlines the bias concerns and helps contextualize the definition and usage of $s$ at the outset.
>
> ## Clarifying the Multi-Attribute Setting
> We acknowledge the confusion surrounding the multiple-attribute case. The original manuscript insufficiently disambiguated the treatment between binary and multi-attribute scenarios, especially around ALA-BA.
>
> The revised manuscript will clarify:
> - In the binary attribute setting (e.g., gender), $s$ denotes the desired attribute as predicted from the image. The text classifier then evaluates whether the generated caption matches this target (e.g., gender match).
> - In the multi-attribute setting (e.g., race and physical appearance, or intersectional bias), the concern is not misclassification of the sensitive attribute in text but rather increased toxicity associated with particular groups. In this case, $s = -1$ indicates a non-toxic target as determined by a toxicity classifier.
>
> This design is motivated by existing literature showing that representational harms in VLMs for multiple-attribute cases manifest more prominently through toxicity, not mislabeling [1]. The setup aligns with fairness literature such as COMPAS, where the goal is to equalize harmful outputs across sensitive groups.
>
> We will reorganize the experimental sections (especially Sections 3 and 4) to clarify how $s$ is defined and used across binary and multi-attribute cases.
>
> [1] Uncovering Bias in Large Vision-Language Models with Counterfactuals
>
> ## On Defining Sensitive Attributes
> We respectfully suggest that the broader question of "who defines what is harmful" falls outside the primary scope of this paper. Our work aligns with the NeurIPS's research scope on "Social and economic aspects of machine learning", which encourages research addressing fairness, bias, and societal impact through technical approaches.
>
> Specifically:
> - The choice of sensitive attributes in our experiments (e.g., gender, race, appearance) follows established norms in fairness literature and is supported by recent studies showing consistent disparities in VLM and LMM outputs.
> - Our contribution lies in proposing a modular, post hoc debiasing framework that is not tied to specific attribute definitions. Instead, it enables researchers and practitioners to plug in relevant sensitive-attribute classifiers based on their ethical and application-specific criteria.
> We do not claim that gender or race are the only valid concerns; they are chosen as representative examples to demonstrate the feasibility and effectiveness of our method. The framework is extensible to other sensitive attributes and domains (e.g., medical imaging, autonomous vehicles) as long as appropriate data and classifiers are available.
>
> By offering a generalizable mechanism rather than prescribing a universal notion of harm, our work aims to empower responsible AI development within the diverse ethical contexts acknowledged by the NeurIPS community.

---

> ### Author Response · Authors · 2025-08-06
>
> ## On Bias in Attribute Classifiers
>
> We respectfully emphasize that our original claim remains correct and is consistent with established fairness literature. We did **not** argue for a causal relationship between target labels (e.g., appearance, profession, etc.) and sensitive attributes (e.g., gender, race). Rather, we acknowledged that **spurious correlations** can exist in either direction and that such correlations can induce biased predictions, which is a core concern in fairness research.
>
> We also agree with the reviewer's point that *even classifiers trained to predict sensitive attributes may carry bias*, especially if trained on unbalanced or confounded data. However, this general possibility does not invalidate the use of such classifiers. What matters is whether their outputs are sufficiently accurate and robust for the specific task at hand.
>
> To mitigate these concerns, we took the following steps:
>
> - **Data selection**: We use the FairFace dataset, which is designed to offer demographic balance and only includes **facial** images, thereby reducing confounding factors like clothing or background.
> - **Empirical validation**: We trained simple logistic regression classifiers on top of frozen image embeddings and evaluated both **classification accuracy** and **uncertainty** (defined as the proportion of ambiguous samples among misclassifications, where $|s| \leq 0.3$):
>
>   | Frozen Encoder | Accuracy | Portion Ambiguous in Misclassified Samples |
>   |----------------|----------|---------------------------------------------|
>   | LLaVA-7B       | 0.9637   | 0.7839                                      |
>   | PaliGemma      | 0.9323   | 0.8302                                      |
>
>   These results demonstrate that the classifier is both accurate and expresses meaningful uncertainty, supporting its use in our framework.
>
> - **Design robustness**: Our ALA-N (Neutralization) variant is explicitly designed to **avoid reliance on the attribute classifier** by setting $s = 0$, making it resilient to imperfect predictions.
>
> Moreover, using sensitive attribute classifiers is a common and accepted practice in debiasing research [2-5]. Their role is to provide consistent signals for alignment or neutralization similar to ALA.
>
> Finally, we emphasize again that our claims are grounded in established fairness definitions and frameworks, including works [5-13]. We believe that our use of attribute classifiers is both theoretically justified and empirically supported, and our framework offers a practical path forward even when perfect data or classifiers are unavailable.
>
>
> [2] Madra et al. Learning adversarially fair and transferable representations. ICML 2018.
> [3] Ramaswamy et al. Fair attribute classification through latent space de-biasing. CVPR 2021.
> [4] Wang et al. Fairness-aware adversarial perturbation towards bias mitigation for deployed deep models. CVPR 2022.
> [5] Jung et al. A Unified Debiasing Approach for Vision-Language Models across Modalities and Tasks. NeurIPS 2024.
> [6] Seth et. al. Dear: Debiasing vision-language models with additive residuals. CVPR 2023
> [7] Lahoti et. al. Fairness without demographics through adversarially reweighted learning. NeurIPS 2020.
> [8] Chai et. al. Fairness without demographics through knowledge distillation. NeurIPS 2022.
> [9] Jung et. al. Adversarial Latent Feature Augmentation for Fairness. ICLR 2025.
> [10] Cho et. al. Dall-eval: Probing the reasoning skills and social biases of text-to-image generation models. ICCV 2023.
> [11] Dixon et. al. Measuring and mitigating unintended bias in text classification. AAAI/ACM Conference on AI, Ethics, and Society 2018.
> [12] Bolukbasi et. al. Man is to computer programmer as woman is to homemaker? debiasing word embeddings. NeurIPS 2016.
> [13] Dehdashtian et. al. FairerCLIP: Debiasing CLIP's Zero-Shot Predictions using Functions in RKHSs. ICLR 2024.

---

> > ### Comment · Reviewer_cnLN · 2025-08-07
> > **Reply to authors**
> >
> > I appreciate the authors' continued discussions.
> >
> > I acknowledge the authors' POV, and understand that it must be frustrating dealing with a reviewer whose views fundamentally differ from the core ideas of debiasing.
> >
> > I think the authors have done a good job of citing works to justify their claims. While I still fundamentally disagree with the notion of bias classification and the definition of sensitive attributes, I realize such critique is more targeted towards debiasing research as a whole rather than this specific paper. **Thus, while I can't recommend acceptance (in good faith), I will also not fight for rejection during the reviewer/AC discussion phase.**
> >
> > If accepted, I trust that the authors will make sure to clarify the definition of the sensitive attribute signal, $s$. I believe this must be improved for better readability of the paper.

---

### Public Comment · ~Hoin_Jung1 · 2026-07-02

Final accepted version: ICLR 2026 Poster. Please cite the ICLR 2026 version: https://openreview.net/forum?id=u02Tgg4UYg

---

### Note · Authors · 2025-08-13

We thank the reviewers and AC for their insightful and constructive review process. We are very pleased with the positive feedback, and the discussions have allowed us to significantly strengthen the paper and clarify its core contributions. We are confident we have successfully addressed all reviewer concerns.

Below is a summary of the key revisions made during the rebuttal period:

1. **Extensive Qualitative Examples (cnLN, tWX2, 4MDi)**: We acknowledge that the original qualitative results in Figure 1  were not sufficient. In response, we have provided six new, detailed qualitative examples comparing ALA against all baselines. These additions, which we will include in the appendix, illustrate how ALA-BA and ALA-N effectively correct bias and stereotyping where other methods fail.
2. **Superior Fairness-Utility Trade-off (JF4X, 4MDi):** Regarding performance, we clarified that ALA’s key advantage is offering the best trade-off between fairness and utility. While some methods might achieve a slightly higher fairness score by aggressively altering representations, they often do so at the cost of model performance. Our results across all experiments demonstrate that ALA achieves top-tier fairness while uniquely preserving the model's caption and VQA quality, making it a more practical and robust solution.
3. **Validation of the External Classifier (tWX2, cnLN):** In response to concerns about relying on an external classifier, we clarified that this is a common and accepted practice in fairness literature. More importantly, we verified that the classifier's performance is sufficiently high and does not create a reliance issue or performance bottleneck. Our lightweight classifiers are effective without harming ALA's overall performance.
4. **Justification for Problem Formulation (cnLN):** We appreciate the questions regarding our motivation and experimental choices. We have provided a more thorough justification for our problem formulation, grounding our definitions of bias and attribute selection in established fairness literature to demonstrate their direct relevance and soundness.

We are confident these revisions have solidified our paper's contributions. Thank you for your time and consideration.

---

### Decision · Program_Chairs · 2025-09-17

**Decision:**

Reject

**Comment:**

The paper proposes a debiasing method for vision language models. The proposal works directly at the logit level to neutralize the bias.

Authors have addressed questions and improved the paper during the rebuttal phase. Reviewers acknowledged the efforts and how the paper has improved since the initial submission. However, there is a consensus that the paper would still need significant improvements to be ready for acceptance.